# Overshooting the critical threshold for the Greenland ice sheet

Nils Bochow[1,2,3 ✉], Anna Poltronieri[1], Alexander Robinson[3,4,5], Marisa Montoya[5,6], Martin Rypdal[1] & Niklas Boers[3,7,8]

Melting of the Greenland ice sheet (GrIS) in response to anthropogenic global warming poses a severe threat in terms of global sea-level rise (SLR)[1]. Modelling and palaeoclimate evidence suggest that rapidly increasing temperatures in the Arctic can trigger positive feedback mechanisms for the GrIS, leading to self-sustained melting[2–4], and the GrIS has been shown to permit several stable states[5]. Critical transitions are expected when the global mean temperature (GMT) crosses specific thresholds, with substantial hysteresis between the stable states[6]. Here we use two independent ice-sheet models to investigate the impact of different overshoot scenarios with varying peak and convergence temperatures for a broad range of warming and subsequent cooling rates. Our results show that the maximum GMT and the time span of overshooting given GMT targets are critical in determining GrIS stability. We find a threshold GMT between 1.7 °C and 2.3 °C above preindustrial levels for an abrupt ice-sheet loss. GrIS loss can be substantially mitigated, even for maximum GMTs of 6 °C or more above preindustrial levels, if the GMT is subsequently reduced to less than 1.5 °C above preindustrial levels within a few centuries. However, our results also show that even temporarily overshooting the temperature threshold, without a transition to a new ice-sheet state, still leads to a peak in SLR of up to several metres.

Melting of the GrIS has contributed more than 20% to the observed SLR since AD 2002 (ref. 7). Modelling results indicate that the GrIS exhibits several stable states, with critical transitions between them when the GMT exceeds a critical threshold[4,6,8]. With further global warming, a partial to complete loss of the ice sheet is expected, implying an increase of the global sea level by up to 7 m (refs. 3,9). The land-ice contribution to SLR until the year AD 2100 is expected to be in the range of several decimetres, with the GrIS being one of the main contributors[10–12]. As well as the direct impacts on coastal ecosystems and populations, the North Atlantic freshening resulting from a melting GrIS might contribute to a weakening or even destabilization of the Atlantic Meridional Overturning Circulation (AMOC), which would have global-scale impacts, including disruptions of the African and Asian monsoon systems[13–16].

In recent decades, meltwater runoff from the GrIS has accelerated relative to global surface temperatures[17] and there are precursor signals of an impending critical transition detectable in ice cores from the central-western GrIS[18]. There is, therefore, a need to explore the future trajectories of the GrIS under different emission scenarios. Furthermore, it is important to understand what is required to prevent a runaway effect. The so-far insufficient efforts to reduce global emissions make it necessary to investigate scenarios in which we do not achieve current warming targets, such as those defined in the Paris Agreement, by the end of the twenty-first century[19–21]. Different options to remove $CO_2$ from the atmosphere, including carbon capture and storage technologies and large-scale reforestation, could make it possible to maintain such temperature goals in the long term, even if a temporary overshoot occurs[22]. These subsequent efforts to reduce GMTs after AD 2100 could have a substantial mitigating effect because many of the large-scale components of the climate system change slowly compared with the current rate of global warming. In the following, we refer to temporary exceedances of temperature targets or critical temperature thresholds as overshoots and to the equilibrium temperatures that will be reached in the long term as convergence temperatures.

Owing to the effect of inertia, crossing a critical threshold in a dynamical system with several stable states does not necessarily imply that a transition to an alternative state is realized. It is possible to temporarily overshoot the tipping threshold of a system without triggering a transition to a new system state[23]. Thus, the temperature threshold of the GrIS could be surpassed without committing to total mass loss, if later on, yet within a specific time frame, actions are taken that reduce the temperature back under the critical threshold.

The overshoot phenomenon is particularly relevant for the GrIS because the timescales for mass loss are long compared with changes in anthropogenic greenhouse emissions. The separation of timescales could make it possible to reverse ice loss if global surface temperatures decrease sufficiently quickly after an initial overshoot. However, because of the complexity of the ice sheet and the various physical

[1]Department of Mathematics and Statistics, UiT – The Arctic University of Norway, Tromsø, Norway. [2]Physics of Ice, Climate and Earth, Niels Bohr Institute, University of Copenhagen, Copenhagen, Denmark. [3]Potsdam Institute for Climate Impact Research, Potsdam, Germany. [4]Alfred-Wegener-Institut, Helmholtz-Zentrum für Polar- und Meeresforschung, Potsdam, Germany. [5]Department of Earth Science and Astrophysics, Complutense University of Madrid, Madrid, Spain. [6]Instituto de Geociencias, CSIC-UCM, Madrid, Spain. [7]Earth System Modelling, School of Engineering & Design, Technical University of Munich, Munich, Germany. [8]Department of Mathematics and Global Systems Institute, University of Exeter, Exeter, UK. ✉e-mail: nils.bochow@uit.no

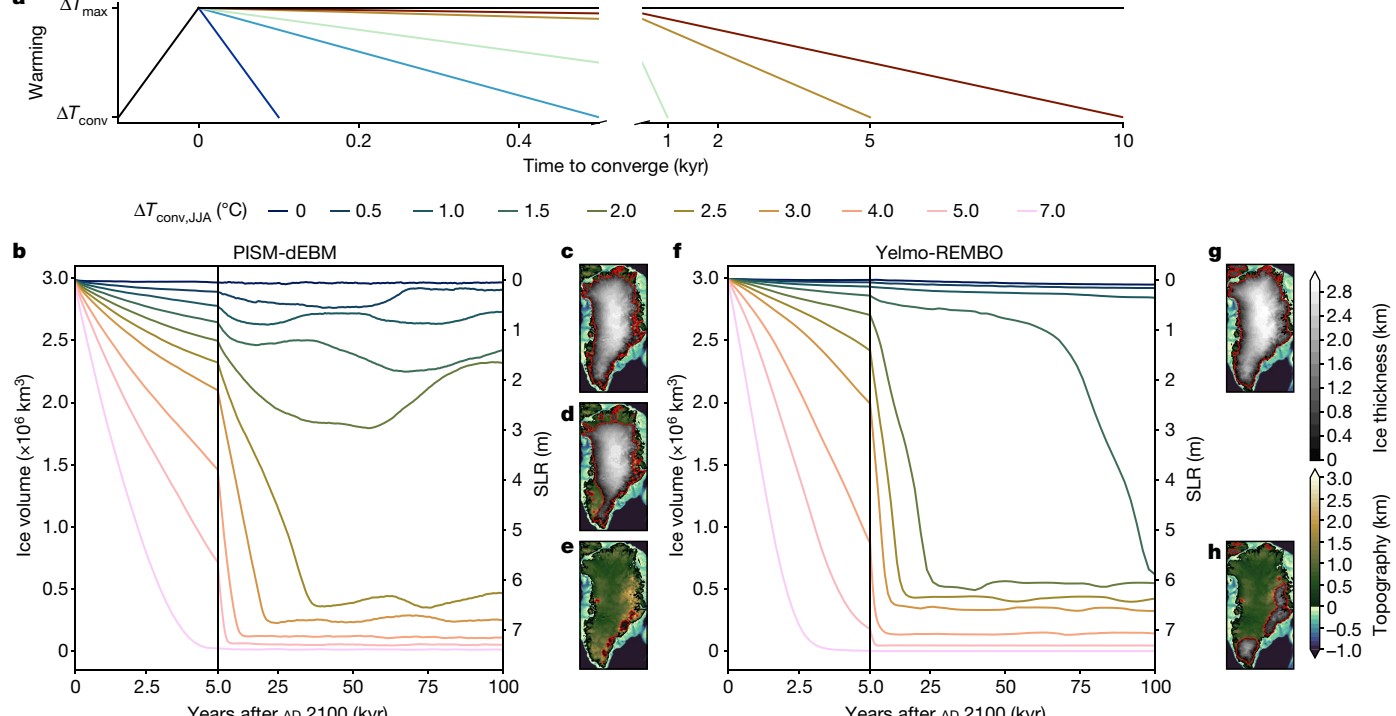

**Fig. 1 | Time series of ice volume and spatial extents of the GrIS for warming scenarios without mitigation. a**, Sketch of applied warming and cooling scenarios in this study. The warming period lasts for 100 years, followed by varying cooling phases. The black line corresponds to scenarios without mitigation as shown in this figure. **b**, Evolution of total GrIS ice volume simulated by PISM-dEBM, without reversal of the temperature anomalies (black line in panel **a**), for different temperature anomalies between $\Delta T_{JJA}$ = 0 °C and 7.0 °C above present. The warming period lasts for 100 years until year AD 2100 and temperatures are kept constant afterwards. Three qualitatively different regimes are noticeable: (1) present-day configuration with fully extended ice sheet or only slightly reduced volume; (2) intermediate state with around 75%

of present-day ice volume; and (3) basically ice-free states. The vertical black line at 5 kyr denotes a change of the x-axis scaling for visual clarity. We normalize the ice volumes to the observed present-day values (see Methods sections 'PISM-dEBM' and 'Yelmo-REMBO'). **c**, Ice thickness of present-day ice-sheet configuration in PISM-dEBM. The ice sheet is fully extended. **d**, Same as **c** but the intermediate state for $\Delta T_{conv,JJA}$ = 2.0 °C, after 100,000 years with PISM-dEBM. The southwestern part of the ice sheet is fully retracted. **e**, Same as **c** but for the ice-free state with PISM-dEBM. **f,g,h**, Same as **b,c,e**, respectively, but for Yelmo-REMBO. Only two regimes can be identified: (1) present-day configuration; and (2) near-ice-free states. The maps were made with the Python package cartopy[52] and Natural Earth.

processes that play a role, including ice flow and surface processes, it is intrinsically challenging to determine the temperature thresholds and required cooling rates that will prevent a substantial loss of the GrIS.

In this modelling study, we identify safe operating spaces by analysing the behaviour of the GrIS under different warming projections that exceed the presumed critical threshold, but in which the temperature is subsequently reduced. We explore the influence of realistic greenhouse gas emission and corresponding warming scenarios for the twenty-first century in accordance with the most recent Intergovernmental Panel on Climate Change report[1]. Subsequently, we apply different idealized carbon-removal scenarios that lead to a temperature decrease on timescales varying from one hundred to tens of thousands of years (Fig. 1a).

We investigate the behaviour of the GrIS using two independent, state-of-the-art ice-sheet models: a new version of the Parallel Ice Sheet Model (PISM) with a modified version of the diurnal Energy Balance Model (dEBM-simple) for the surface mass balance[24,25] and the ice-sheet model Yelmo[26] coupled to the Regional Energy-Moisture Balance Orographic (REMBO) model[27]. Both approaches have been extensively tested and validated and have been used to simulate the past, present-day and future evolution of ice sheets[10,11,25,28–32].

We force the two models, PISM-dEBM-simple (hereafter PISM-dEBM) and Yelmo-REMBO, by a prescribed change in regional summer temperature relative to present day and apply a scaling factor of 1.61 between regional winter and summer surface temperature to obtain the temperature forcing over the seasonal cycle. This forcing can then

be translated into GMT above preindustrial through a linear scaling that accounts for higher warming rates in the Arctic region relative to the global mean (see Methods section 'Climate forcing').

In a first set of experiments, we force the models with a prescribed linear summer (June, July, August (JJA)) temperature increase from year AD 2000 (present day) to AD 2100 to a maximum summer temperature anomaly of $\Delta T_{max,JJA}$ (Fig. 1a). Thereafter, we linearly decrease the temperature between AD 2100 and AD 2200 back to different convergence temperature anomalies between $\Delta T_{conv,JJA}$ = 0 °C and 4.0 °C above present day (that is, $\Delta T_{conv,GMT}$ = 0.5 °C and 3.9 °C convergence GMT above preindustrial (see Methods section 'Climate forcing'). We keep the prescribed temperature anomaly constant after AD 2200 and run the models for another 100 kyr to study the long-term evolution of the ice sheet for each peak warming scenario. In a second set of experiments, we investigate the timescale dependence of the GrIS response following the cooling. After the initial temperature increase until AD 2100, we vary the convergence time ($\Delta t_{conv}$), that is, the time needed to reach the convergence temperature, with $\Delta t_{conv}$ spanning from 100 years to several millennia for various convergence temperatures. We then investigate the behaviour of the GrIS for these different cooling scenarios.

## Evolution without long-term temperature reductions

When kept constant after year AD 2100, the temperature increase during the twenty-first century leads to at least some further melting of

the GrIS for every prescribed positive temperature anomaly (Fig. 1b,f). However, the melt is moderate for temperature anomalies smaller than 1.0 °C for both models. In the long term, the runs with PISM-dEBM show that there is a substantial ice-volume loss of more than 20% for $\Delta T_{JJA} > 1.0$ °C and more than 80% loss for $\Delta T_{JJA} > 2.2$ °C (Fig. 1b). In runs with Yelmo-REMBO, a temperature anomaly $\Delta T_{JJA} > 1.4$ °C leads to a complete melting of the ice sheet (Fig. 1f). Yelmo-REMBO only has two stable ice-sheet states: a close to present-day state and a near-ice-free state (Fig. 1g,h). For PISM-dEBM, there is an extra regime; several intermediate states with around 50–90% of current GrIS ice volume are accessible (Fig. 1c–e). The spatial extent of the different ice-sheet states is in accordance with earlier work[3,5,33].

The intermediate states in the runs with PISM-dEBM show a gradual and eventual complete retreat of the southwestern part of the ice sheet (Extended Data Fig. 1). Simultaneously, there is a retreat of the ice sheet in the northern part of the GrIS, yet the southwestern part is the most sensitive to warming. For a warming $\Delta T_{JJA} > 2.2$ °C the remaining GrIS is lost abruptly. The ice sheet fluctuates on a decamillennial timescale for some configurations and does not reach a steady state. For a warming of $\Delta T_{JJA} = 2.0$ °C, the ice sheet recovers back to approximately 75% of the present-day ice-sheet volume after an initial loss of 40% of the ice-sheet volume (Fig. 1d). The recovery is a result of the glacial isostatic adjustment[34]. The uplift of the bedrock counteracts the melt-elevation feedback and leads to colder temperatures, which allow the ice sheet to partially regrow[34]. Although the same simulations with Yelmo-REMBO do not show any stable intermediate states, the ice sheet does show the same spatial sensitivity to warming, with an initial retreat of the southwestern GrIS followed by a retreat of the northern part of the ice sheet (Extended Data Fig. 2). For the most extreme warming scenario of $\Delta T_{JJA} = 7.0$ °C, the ice sheet is lost in less than 5,000 years in both models.

## Short-term overshoots

A reduction in temperature from AD 2100 to AD 2200 leads to a mitigation of the ice loss, depending on the convergence temperature reached (Fig. 2). Regardless of the peak temperature in AD 2100, a convergence temperature increase of 1.5 °C GMT above preindustrial ($\Delta T_{JJA} = 1.3$ °C) by AD 2200 or lower leads to a stable ice sheet, with the equivalent of less than 1 m long-term SLR contribution in simulations with both models (Fig. 2a,b). However, the maximum interim SLR contribution with PISM-dEBM slightly exceeds 1 m for 1.5 °C GMT above preindustrial (Extended Data Fig. 3). For convergence temperatures $\Delta T_{JJA} > 2.2$ °C for PISM-dEBM and $\Delta T_{JJA} > 1.4$ °C for Yelmo-REMBO, the ice sheet is completely lost, regardless of the overshoot temperature in the year AD 2100. The safe zone is sharply separated from the transition area, which is visible as an abrupt transition in the cross-sections of the stability diagram (Fig. 2c,d). Although the ice sheet shows a more gradual loss before the critical threshold with PISM-dEBM (Fig. 2c), the ice loss is more abrupt with Yelmo-REMBO and the SLR contribution is less than 1 m before the critical threshold is crossed (Fig. 2d). Regardless of the model, the ice-sheet equilibrium only depends on the absolute temperature increase by AD 2200, that is, the convergence temperature anomaly, and not the peak value at AD 2100. This can be explained by the slow response timescale of the ice sheet to the temperature change.

For low convergence temperature anomalies, the ice-sheet volume barely changes in simulations with either model. For high warming, the equilibration time is very slow, on the timescale of decamillennia. For intermediate warming levels, the ice sheet does not reach a classical equilibrium in simulations with PISM-dEBM but fluctuates on decamillennial timescales. This is particularly true for the intermediate states close to the threshold of $\Delta T_{JJA} = 2.2$ °C, which are not in equilibrium after even 100 kyr (triangle symbols in Fig. 2a). Likewise, the simulations with Yelmo-REMBO forced with $\Delta T_{conv,JJA} = 1.5$ °C are not yet in equilibrium after 100 kyr and eventually evolve further towards the ice-free state (Fig. 2b,d).

## Long-term overshoots

To investigate the timescale dependence of the overshoot of the temperature threshold, we decrease the temperature after AD 2100 to different convergence temperatures ranging from $\Delta T_{JJA} = 0$ °C to 4.0 °C and vary the convergence time to reach the respective convergence temperature from 100 years to several millennia (Fig. 1a). All scenarios considered show a loss of ice volume. As expected, the longer the convergence time and the higher the overshoot temperature, the larger the ice loss. However, there are important dependencies of the ice-sheet evolution (and thus maximum SLR contributions) on the exact convergence times and temperatures (Fig. 3). For a convergence time of 1,000 years, the maximum SLR contribution is similar to the equilibrium and maximum SLR contributions for a 100-year convergence time (Fig. 2 and Extended Data Figs. 3 and 4a,b), implying that the maximum ice loss is reached after the warming and cooling phase. However, an overshoot temperature of $\Delta T_{max,JJA} > 6.0$ °C leads to a greater maximum SLR contribution than at equilibrium (Extended Data Figs. 3 and 4a,b). Even for a convergence temperature of $\Delta T_{conv,JJA} = 0$ °C, the maximum SLR contribution exceeds 1 m for the highest overshoot temperature in both models (Fig. 3a,b). For a convergence time of 10,000 years, there is a strong dependence of the maximum SLR contribution on the overshoot temperature (Fig. 3c,d). Both models exceed 1 m SLR contribution for an overshoot temperature $\Delta T_{max,JJA} > 2.5$ °C in the year AD 2100, given a convergence temperature of $\Delta T_{conv,JJA} = 0$ °C. For an overshoot temperature of $\Delta T_{max,JJA} > 6.0$ °C with a subsequent return to present-day conditions, the simulated SLR contribution is at least 5 m with PISM-dEBM and 7 m with Yelmo-REMBO.

For a convergence temperature of $\Delta T_{conv,JJA} = 0$ °C, we find that, for all scenarios, the ice sheet eventually returns to values close to the present-day ice volume in both models (Fig. 4). For the short-term overshoots ($\Delta t_{conv} < 500$ years), the models show very similar SLR contributions and the maximum ice-volume loss before ice-sheet regrowth is in the range of 50 cm SLR equivalent (Fig. 4a,b). For a convergence time of 1,000 years, the SLR contribution is less than 1.25 m with either model, followed by a recovery to the present-day ice sheet. For convergence times of more than 5,000 years, a complete loss of the ice sheet can occur before recovery, with a SLR contribution of 7 m (Fig. 4c,d). Although Yelmo-REMBO shows a complete loss of the ice sheet, before regrowth, for the highest overshoot temperatures and a convergence time of 5,000 years, PISM-dEBM only shows a complete loss, before recovery, for a convergence time of 10,000 years, regardless of the convergence temperature (Fig. 5a,b).

For higher convergence temperatures, the GrIS does not necessarily return to its present-day ice volume, highlighting the potential practical irreversibility caused by the hysteresis of the ice sheet (Fig. 5c,d). With PISM-dEBM, the ice sheet approaches the intermediate states noted above. The ice-volume loss at equilibrium gradually increases with increasing convergence temperature, reaching up to 25% of the present-day ice volume for a convergence temperature of $\Delta T_{conv,JJA} = 2.2$ °C. However, with PISM-dEBM, the ice sheet always recovers to the equivalent equilibrium, as for a simple ramp-up simulation (which we will refer to as the reference simulation hereafter; black lines in Fig. 5) for a given temperature anomaly. In simulations with Yelmo-REMBO, the ice sheet does not always regrow to the same ice volume corresponding to the reference simulation (Fig. 5d). Close to the threshold, the ice sheet shows a dependence on the convergence time. A convergence time greater than 5,000 years, combined with a high overshoot temperature, prevents regrowth of the ice sheet even below the critical threshold (Extended Data Fig. 4d). For a convergence temperature of $\Delta T_{conv,JJA} = 0.5$ °C and long convergence times, the ice sheet regrows to an intermediate state with around 2 m SLR contribution after a complete loss (Fig. 5d).

For all scenarios, the maximum SLR contribution strongly depends on the maximum temperature, the convergence temperature and the

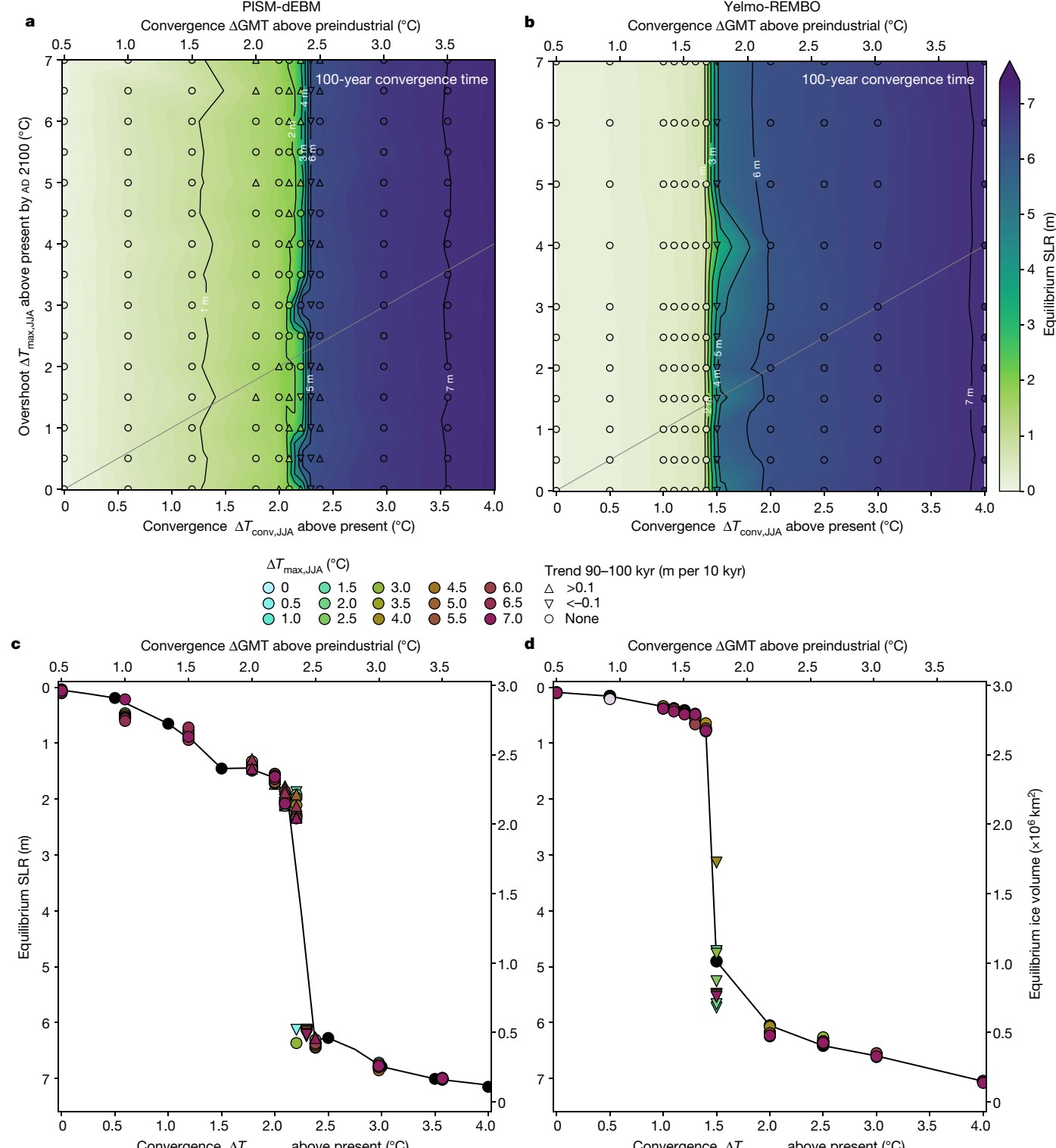

**Fig. 2 | Stability diagram of the GrIS after warming and subsequent cooling. a**, Stability diagram of the GrIS for PISM-dEBM. Different warming rates are applied for 100 years, followed by various cooling rates for another 100 years. The temperature is kept constant afterwards for another 100 kyr. White regions indicate a present-day-like ice sheet, green–blue regions mark intermediate states and purple corresponds to the ice-free state. The grey line corresponds to the warming rates at which the overshoot temperature equals the convergence temperature (that is, no mitigation; the time series of simulations along the grey line is depicted in Fig. 1). Below the grey line,

the overshoot temperature in year AD 2100 is smaller than the convergence temperature in AD 2200. Corresponding time series of every simulation are shown in Extended Data Fig. 5. **b**, Same as **a** but for Yelmo-REMBO. **c**, Cross-sections of the stability diagram for all applied overshoot temperatures indicated on the $y$ axis of **a**. A sharp decrease of the ice volume can be inferred for $\Delta T_{\mathrm{conv,JJA}}$ above 2.2 °C in all cross-sections, resulting in several intermediate and ice-free GrIS states. **d**, Same as **c** but for Yelmo-REMBO, for which the critical temperature is around $\Delta T_{\mathrm{conv,JJA}} = 1.4$ °C. The triangles mark simulations that have still not converged during the time span from 90 kyr to 100 kyr (see legend).

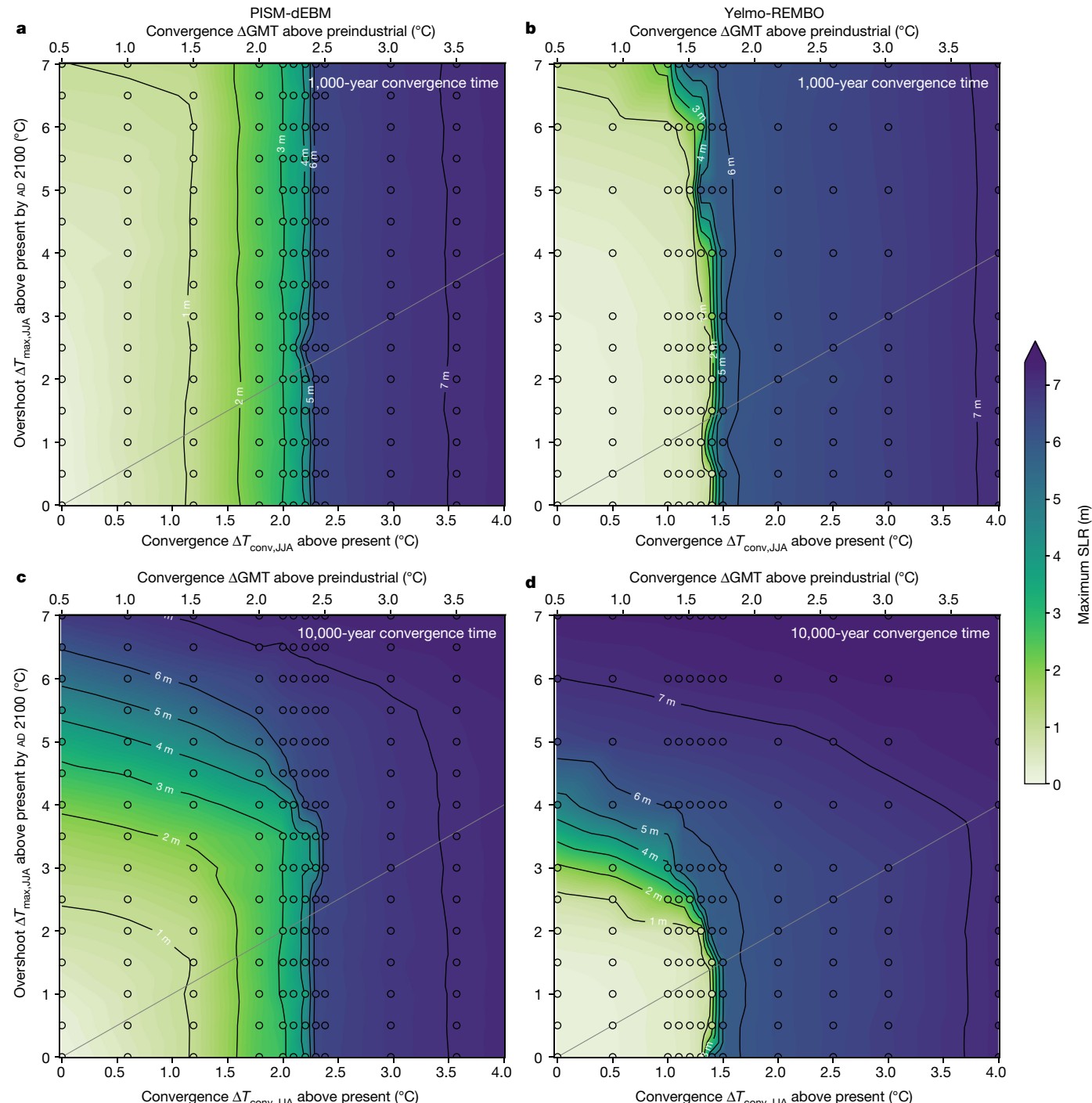

**Fig. 3 | Maximum SLR contribution of the GrIS after warming and subsequent cooling for two different convergence times. a**, Maximum SLR contribution of the GrIS for PISM-dEBM, for 1,000 years convergence time. Different warming rates are applied for 100 years, followed by various cooling rates for a convergence time of 1,000 years. The temperature is kept constant afterwards for another 100 kyr. **b**, Same as **a** but for Yelmo-REMBO. **c,d**, Same as

**a,b**, respectively, but for a convergence time of 10,000 years. The maximum SLR contribution shows a clear dependence on the overshoot temperature. White regions indicate a present-day-like ice sheet, green–blue regions mark intermediate states and purple corresponds to the near-ice-free state. The grey lines correspond to the scenarios for which the overshoot temperature equals the convergence temperature.

convergence time (Fig. 5). Generally, the larger the maximum temperature, the convergence time and the convergence temperature, the larger the maximum SLR contribution. The longer the convergence times, the stronger the dependence of the maximum SLR contribution on the overshoot temperature (Fig. 6). Our key result is that, regardless of the model used, it is possible to define safe and unsafe scenarios dependent on a chosen target maximum SLR contribution.

For example, we find that a convergence time shorter than 1,000 years with a convergence temperature around $\Delta T_{conv,JJA} = 0$ °C keeps the GrIS SLR contribution below 2 m for all overshoot temperatures (Fig. 6) with both models. For overshoot temperatures below the critical threshold, the maximum SLR contribution is weakly dependent on the convergence time, which is not surprising given that the maximum SLR contribution for a given maximum temperature anomaly

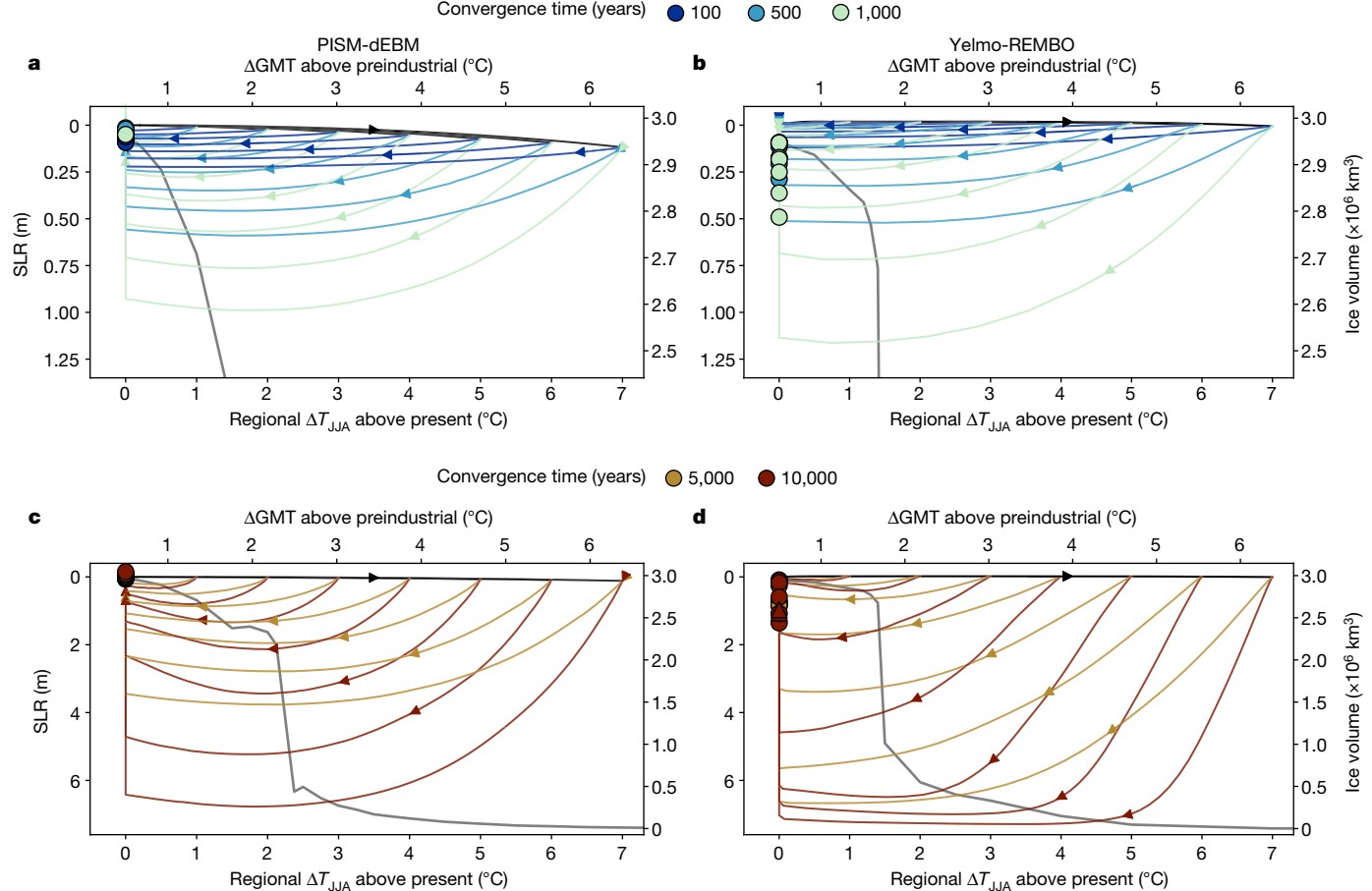

Convergence time (years) ● 100 ● 500 ○ 1,000

**a** PISM-dEBM
ΔGMT above preindustrial (°C)

**b** Yelmo-REMBO
ΔGMT above preindustrial (°C)

Convergence time (years) ● 5,000 ● 10,000

**c**
ΔGMT above preindustrial (°C)

**d**
ΔGMT above preindustrial (°C)

**Fig. 4 | Trajectories of overshoot scenarios converging to a regional summer temperature of 0 °C above present (0.5 °C GMT above preindustrial level) for various peak warmings and convergence times. a**, Trajectories of ice-sheet volume for PISM-dEBM for convergence times of 100, 500 and 1,000 years. All three scenarios show an ice loss that reaches its maximum during the cooling phase. The apparent jump of the end states (dots) at $\Delta T_{JJA} = 0$ °C corresponds to a recovery of the ice sheet after the cooling phase. The end states are defined as the mean ice volume after 90–100 kyr. The thick

dark grey line corresponds to the equilibrium states for the applied temperature anomaly, showing that the actual, realistic trajectories are strongly out of equilibrium. **b**, Same as **a** but for the ice-sheet model Yelmo-REMBO. **c,d**, Same as **a,b**, respectively, but for convergence times of 5,000 and 10,000 years. For all scenarios, both models show a recovery to close to the present-day ice sheet. The maximum SLR contribution is reached during the cooling phase, highlighting the importance of considering long-term committed SLR in climate negotiations.

is generally equal to or lower than the equilibrium SLR contribution of that forcing value (Fig. 3).

## Discussion

We use two different state-of-the-art ice-sheet-modelling approaches, with varying complexity, and show that the results obtained from both approaches are consistent, despite the fact that the feedbacks captured by the models differ to some extent. We use a recently published version of PISM that is driven at the surface by the dEBM (PISM-dEBM) to capture surface albedo feedbacks. This improves on the more conventional positive degree-day parameterization, which might fail for past and future climate conditions[35–39]. Increased surface melt reduces reflectivity of the ice-sheet surface and hence leads to an increase in the melt rates, which is captured by the dEBM. Although the extra atmospheric warming that can result from reducing albedo is not captured by this model setup, Yelmo-REMBO includes this feedback as the atmosphere is dynamically coupled to the snowpack energy balance. Possible negative atmospheric feedbacks that have been shown to potentially decelerate the ice loss are also not included in PISM-dEBM. It has been shown that changes in cloud cover, circulation patterns and precipitation lead to increased accumulation in the high-altitude, cold interior of the ice sheet and can increase the critical temperature threshold[5]. However,

Yelmo-REMBO includes a dynamic albeit simple atmosphere that produces increased precipitation following the retreating ice-sheet margin and therefore captures the negative feedbacks at least to some degree. Nevertheless, we propose to extend the work presented here to a setup with a fully coupled, comprehensive atmosphere general circulation model as an interesting follow-up study.

It has recently been shown that, to some extent, glacial isostatic adjustment can counteract the positive feedbacks that are believed to cause a hysteresis of the GrIS with global warming, such as the melt-elevation feedback and albedo feedback[34]. However, the timescale of this feedback is still debated[40,41] and is often neglected on sub-millennial timescales[3]. The fluctuations of the ice sheet on a decamillennial timescale simulated by PISM-dEBM are believed to be the consequence of an interplay between bedrock uplift and melt-elevation feedback[34,42]. We find that the intermediate GrIS states found with PISM-dEBM are at least partially caused by the interplay between the glacial isostatic adjustment and melt-elevation feedback and we find fewer intermediate states without bedrock uplifting (Extended Data Fig. 6e). Palaeoclimatic simulations of the Pliocene GrIS show similar intermediate states as seen with PISM-dEBM[42]. By strong contrast, however, Yelmo-REMBO uses the same Earth deformation model and we do not observe similar oscillations with this model. This may point to a different balance between positive feedbacks (largely

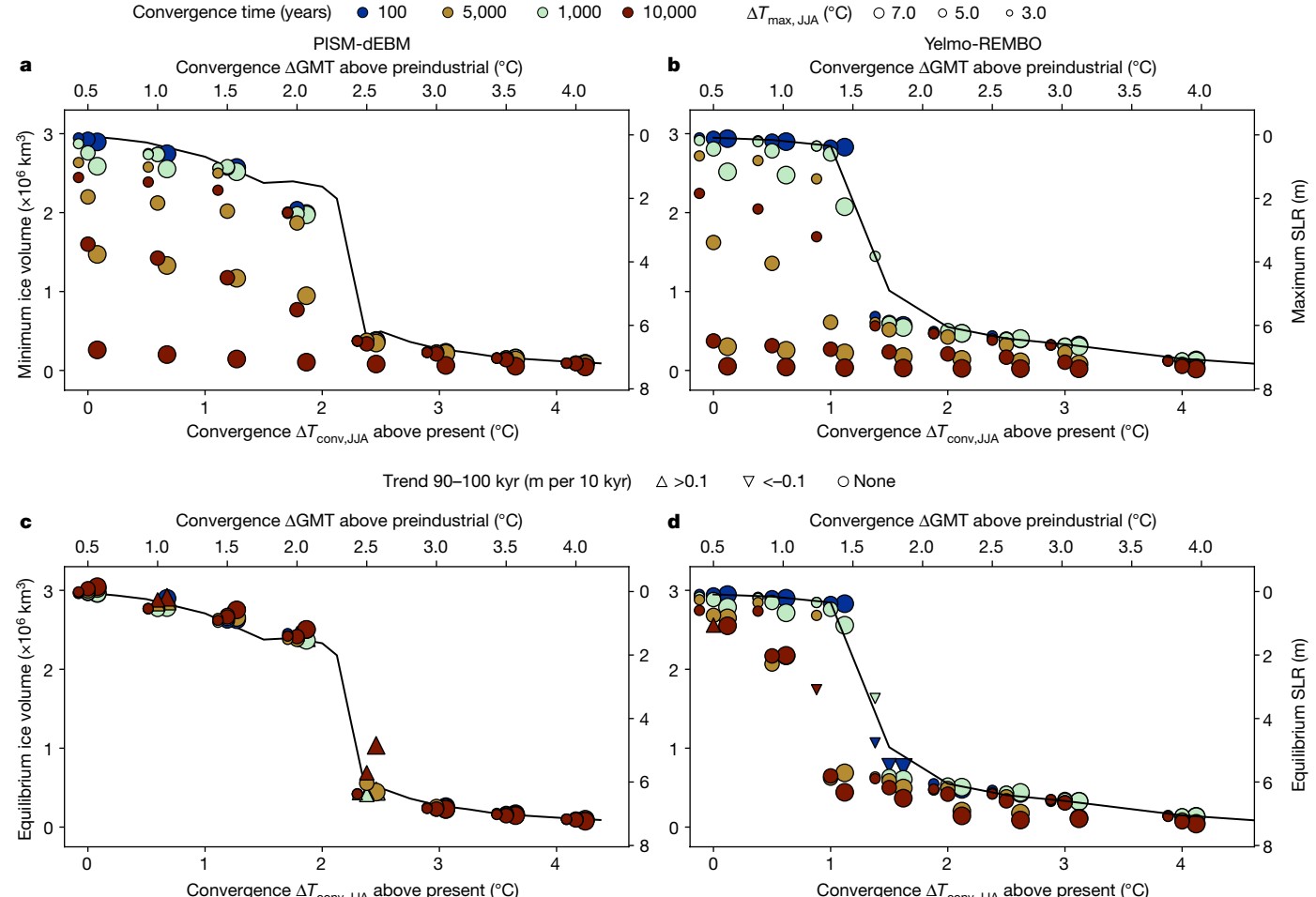

**Fig. 5 | Minimum and equilibrium ice volume for different overshoot scenarios. a**, Minimum ice volume and maximum SLR contribution for different convergence temperatures ($\Delta T_{\mathrm{conv,JJA}}$) between 0 °C and 4.0 °C above present, overshoot temperatures between $\Delta T_{\mathrm{max,JJA}}$ = 3.0 °C and 7.0 °C and convergence times between 100 and 10,000 years for PISM-dEBM. For higher overshoot temperatures and longer convergence times, the minimum ice volume is lower. A convergence time of 10,000 years leads to a complete, temporary loss of the GrIS for all overshoot temperatures. The black line corresponds to the equilibrium reference simulation without any

temperature decrease. **b**, Same as **a** but for Yelmo-REMBO. The behaviour is similar to PISM-dEBM except for the fact that a complete temporary GrIS loss is already possible for shorter convergence times of 5,000 years. **c,d**, Same as **a,b**, respectively, but for the ice volume after 90–100 kyr. The triangles denote simulations that still show a trend after 100 kyr. The ice sheet regrows to the reference simulation in all cases with PISM-dEBM but not with Yelmo-REMBO. The latter shows a temperature range of roughly 0.5 °C below the critical threshold, which shows irreversibility after a complete loss of the GrIS.

at the surface) and the glacial isostatic rebound and should certainly be studied with more models in future work.

Our temperature thresholds are in accordance with previous work[4,6,8,43–45] and agree with the general consensus that limiting global warming below the range of 1.5–2.5 °C above preindustrial levels can prevent the most severe consequences[6,8]. However, we do not aim to give a precise threshold value for the safe zone but rather to show that it is possible to mitigate a critical loss of the GrIS and the associated SLR contribution if efforts are made to (1) prevent extreme warming by AD 2100 and (2) reduce the temperature after a reasonable time, that is, centuries. Failing in either of these efforts can result in large SLR contributions from the GrIS even for convergence temperatures of between 0 and 1.5 °C above preindustrial.

Notably, in the warming-only experiments, we find that several intermediate stable states of the GrIS are accessible with PISM-dEBM as temperatures increase before the remaining ice sheet is lost abruptly, but not with Yelmo-REMBO. This seems, therefore, to be a model-dependent behaviour that is a result of applying different ice dynamics, climatic forcing and interactions within the system. It is clear that the existence of the intermediate states facilitates reversibility of

the ice loss before the final threshold is crossed with PISM-dEBM. In previous studies that investigate the short-term response of the GrIS to global warming, it has been shown that future projections can differ substantially across models[10,11]. Yet, we find qualitatively remarkably similar behaviour with both models used here. A coordinated model intercomparison following an experimental setup such as the one used here would help to constrain the uncertainty in potential critical thresholds and the long-term future ice-sheet evolution.

Our simulations are restricted to horizontal resolutions of 16–20 km, which means that small-scale processes are not well represented. The choice of this resolution was because of computational constraints and the large number of simulations. However, we are mostly interested in the large-scale evolution of the GrIS on decamillennial timescales. Previous work has shown that the chosen resolutions give similar results to higher-spatial-resolution simulations[3], so we expect that our conclusions are robust. Nonetheless, this should be a target for future work.

Long-term climate projections for Greenland remain uncertain, as most Earth-system-model simulations typically end by the year AD 2100 (ref. 46). Although we based our estimate of Arctic amplification on Coupled Model Intercomparison Project (CMIP) Phase 6 (CMIP6)

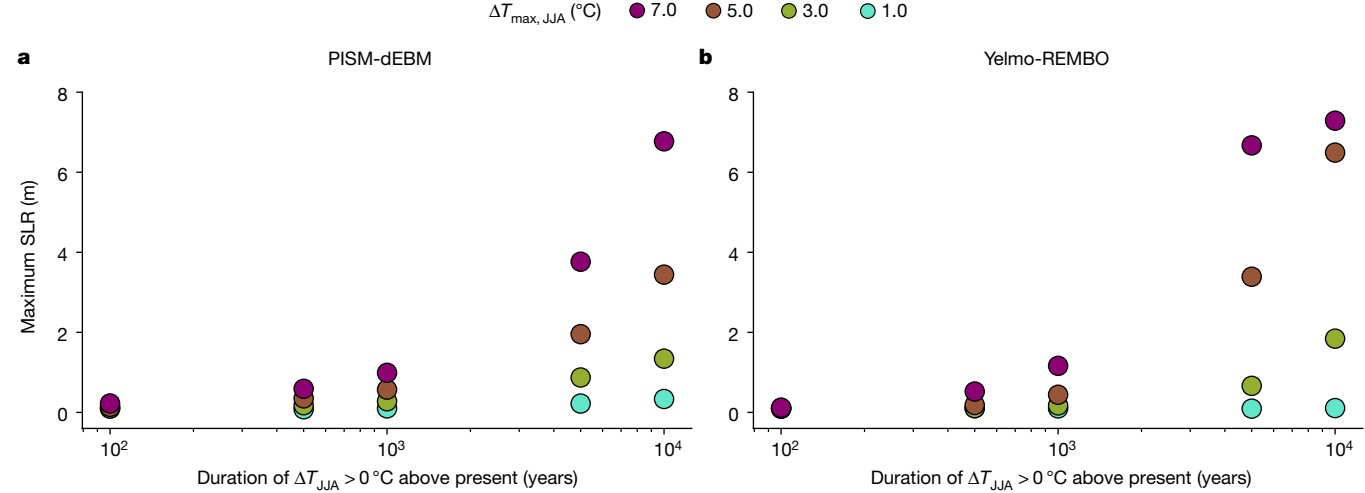

**Fig. 6 | Maximum SLR contribution for different overshoot scenarios with a convergence GMT of 0.5 °C above preindustrial (corresponding to $\Delta T_{JJA}$ = 0 °C above present). a**, Maximum SLR contribution for four different overshoot temperatures and convergence times, up to 10,000 years. **b**, Same as **a** but for Yelmo-REMBO. On timescales of less than 1,000 years, the models show a maximum SLR contribution of less than 2 m for all overshoot temperatures. An overshoot temperature of less than 3 °C prevents a SLR contribution of more than 2 m. On long timescales, Yelmo-REMBO shows a slightly higher SLR contribution for high overshoot temperatures than PISM-dEBM.

models, there is considerable uncertainty about the extent of future warming in the Arctic. Recently, it has been shown that the Arctic warms four times faster than the global average and thus substantially exceeds previous estimates and projections from climate models[47]. Arctic amplification of this magnitude would reduce the safe space for the GrIS substantially. However, surface temperatures around Greenland might not increase that severely in the future[47,48]. On multimillennial timescales, there may be substantial changes in global climate, atmosphere and ocean circulation that are hard to quantify today. For example, a weakening AMOC leads to decreasing Greenland temperatures[13,49], which could help to restabilize the ice sheet. However, at the same time, a weakening of the AMOC is expected to decrease precipitation over Greenland[13,49], which could lead to the opposite effect and destabilize the GrIS even more. These further interactions should be tackled in the future by Earth system models with interactive ice-sheet components.

The potential irreversibility of a loss of the GrIS is an important concern[8,50]. Our results show that mitigation of an ice-sheet loss is possible if temperatures are reduced relatively quickly after a temporary overshoot. We find several stable intermediate ice-sheet configurations with PISM-dEBM that return to the present-day state if the climate returns to present-day conditions. However, if longer time spans are needed to cool down to a relatively safe convergence GMT of, for example, 1.0 °C, the SLR contribution from the GrIS can still exceed several metres for thousands of years. With Yelmo-REMBO, there is a temperature range of 0.5 °C below the threshold that shows irreversibility; even if the convergence temperature is below the critical threshold after an initial overshoot, the GrIS does not regrow. This emphasizes the risk of an irreversible ice-sheet loss for long-term overshoot scenarios. Moreover, total runoff amounts would still be substantial even for a reversible ice-sheet loss, with possibly severe consequences for the AMOC[51]. Remarkably, the timescale of ice loss relative to their respective thresholds agrees very well across the two models used here. It should be emphasized nevertheless that quantitative differences between the two ice-sheet models are present and should be investigated in the future.

We find a threshold for an abrupt, complete loss of the GrIS around 2.3 °C GMT above preindustrial level with PISM-dEBM and 1.7 °C GMT above preindustrial level with Yelmo-REMBO, which is in agreement with previously reported critical temperatures for the GrIS[4,6,43–45]. We show that a transition to an ice-free GrIS state can be avoided in scenarios that overshoot this critical temperature threshold, as long as the temperature anomaly is subsequently reduced sufficiently quickly. Our results highlight the critical role of warming and cooling rates as well as the maximum and convergence temperatures. In our simulations, southwestern Greenland is most sensitive to temperature changes and primarily determines the spatial extent of the potential intermediate states. However, even without an irreversible transition to a new stable ice-sheet state, the intermediate SLR contribution from the GrIS can exceed several metres, depending on the warming and cooling rate, as well on as the convergence temperature.

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

# Methods

## PISM-dEBM-simple

We use the open-source, state-of-the-art PISM version v1.2-41-g53a9818 with the dEBM-simple surface mass balance module and parameterized climate forcing. PISM is a three-dimensional, thermomechanically coupled ice-sheet/ice-shelf model that combines the shallow-ice approximation (SIA) and shallow-shelf approximation (SSA) of the non-Newtonian Stokes model. This hybrid SSA + SIA approach permits modelling of the whole domain from the ice-sheet flow zone with grounded ice to the ice-shelf flow zones in an appropriate manner[24]. The ice rheology is based on the Glen–Paterson–Budd–Lliboutry–Duval flow law[53] with an exponent of $n = 3$ with the enhancement factors $E_{SSA} = 1$ and $E_{SIA} = 3$ for the SSA and SIA flow, respectively.

We use a pseudo-plastic sliding law[54] of the form

$$\boldsymbol{\tau}_b = -\tau_c \frac{\mathbf{u}}{u_0^q |\mathbf{u}|^{1-q}},$$

with the basal shear stress $\boldsymbol{\tau}_b$, basal sliding velocity $\mathbf{u}$, yield stress $\tau_c$ and a threshold velocity $u_0$. We chose $q = 0.5$ and a threshold velocity of $u_0 = 100$ m year$^{-1}$ for our simulations.

The yield stress is determined by the Mohr–Coulomb criterion[55]

$$\tau_c = c_0 + (\tan\phi)\, N_{till}$$

that connects the effective pressure $N_{till}$, a material property field $\phi$ (till friction angle) and the till cohesion $c_0$. The effective pressure $N_{till}$ is determined by the subglacial hydrology model, the till friction angle $\phi$ is a piecewise linear function of bed elevation[56] and the till cohesion $c_0$ is set to 0.

We model the deformation of the Earth owing to the changes in the ice load using the Lingle–Clark model[57,58]. The model is described by a purely elastic lithosphere with a flexural rigidity of $5 \times 10^{24}$ N m$^{-1}$ and the upper mantle is represented as a three-dimensional viscous half-space with a viscosity of $10^{21}$ Pa s$^{-1}$. The model uses a time-dependent partial differential equation that generalizes and improves on the standard elastic plate lithosphere model (ELRA)[58].

To calculate the surface mass balance, we use a recently developed dEBM-simple[25]. The dEBM-simple is a modified version of the earlier introduced full dEBM[37,39]. We use the standard parameters used by Zeitz et al.[25], except for the coefficients $c_1$ and $c_2$, which calibrate the energy balance of the snowpack in the melt equation. These we set to $c_1 = 20$ W m$^{-2}$ K and $c_2 = -50$ W m$^{-2}$, based on an optimization of the product of temporal and spatial root-mean-square error of the surface mass balance with regards to the MARv3.12 regional climate model surface mass balance from 1980 to 2000 (ref. 59). We keep the orbital parameters fixed to the present-day values[25]. The transmissivity of the atmosphere is given by a linear function and assumed not to change in future climate. For an extensive description of the dEBM and the implementation in PISM, see refs. 25,37,39.

The present-day near-surface temperature and precipitation rates are given by climatological means (monthly 1980–2000) from the regional climate model MARv3.12 (ref. 59). We apply an elevation-dependent correction of the surface temperature and precipitation, imposing a lapse rate of $\Gamma = 6$ K km$^{-1}$. The precipitation $P$ changes 3.6% per degree of temperature change. The change of precipitation with increasing temperature is derived from a linear fit of the mean annual precipitation against surface air temperature from 37 CMIP6 SSP585 runs (Extended Data Table 2). We use the default spatiotemporal constant ocean boundary conditions with a constant sub-shelf melt rate of 0.05 m year$^{-1}$.

Our simulations are initialized from a reference equilibrium state of the GrIS that resembles the present-day configuration. We show the ice-surface elevation and ice-surface velocity deviation from observational data in Extended Data Fig. 7. To obtain our reference state, we bootstrap the ice-sheet model from present-day conditions, including ice thickness and bedrock elevation, taken from BedMachine v5 (refs. 56,60), and basal heat flux[61], as well as climatological mean (monthly 1980–2000) surface temperature and precipitation taken from the regional climate model MARv3.12 (ref. 59). We run the model until an equilibrium state is reached, but for at least 50,000 years. All simulations were performed on a regular rectangular grid with a horizontal resolution of 20 km and an equally spaced grid in the vertical direction with a resolution of 40 m.

We normalize the ice volume such that the initial volume corresponds to the observed ice volume of 7.42 m sea-level equivalent in all plots[56].

## Yelmo-REMBO

The ice-sheet model Yelmo[26] resolves ice dynamics by means of the higher-order DIVA solver[62]. Thermodynamics are linked to dynamics by means of effective viscosity, which is determined with a Glen's flow law formulation ($n = 3$) and enhancement factors in the shearing, streaming and floating regimes of 3, 1 and 0.7, respectively. The basal friction is determined with a regularized Coulomb law[63] of the form

$$\boldsymbol{\tau}_b = -c_b \left( \frac{|\mathbf{u}_b|}{|\mathbf{u}_b| + u_0} \right)^q \frac{\mathbf{u}_b}{|\mathbf{u}_b|},$$

with $u_0 = 100$ m year$^{-1}$ and $q = 0.2$. $c_b = c_0 + (\tan\phi) N_{till}$ is the basal yield stress (Pa), in which $N_{till}$ is the effective pressure at the base and $\phi$ represents the material strength of the bed as a till friction angle. As in PISM, $c_0 = 0$ and $\phi$ is set as a piecewise linear function of bedrock elevation with $\phi_{min} = 0.5°$ at bedrock elevations at or below $-700$ m and $\phi_{max} = 40°$ at or above 700 m. Effective pressure at the base of the ice sheet is modelled following ref. 64. When ice is frozen at the base, then the effective pressure equals the overburden pressure ($N_{till} = \rho g H$), and when a saturated water layer is present for temperate ice, the effective pressure reduces to 2% of the overburden pressure value. To determine the basal water layer thickness, basal hydrology is resolved locally (no horizontal transport), depending on water production from melting/freezing the base of the ice sheet and a constant till drainage rate of 1 mm year$^{-1}$. The water layer is limited to 2 m, at which point the till below the ice sheet is considered saturated. Geothermal heat flux is imposed using the reconstruction in ref. 61. Glacial isostatic adjustment of the bedrock is determined using the Lingle–Clark model, as with PISM, and the same parameter values are used. Yelmo is run at 16-km horizontal resolution, with ten terrain-following coordinates in the vertical dimension. The ice-sheet model is coupled bidirectionally to the regional climate model REMBO[27]. REMBO is a two-dimensional energy–moisture balance model in the atmosphere. At the ice-sheet surface, the snowpack is modelled as a single layer. The surface energy balance is approximated through the insolation–temperature melt equation, which accounts for changes in insolation and temperature, as well as surface albedo, but ignores other components. The snowpack and atmosphere evolves with a daily time step over the year and provides the mean annual surface temperature and surface mass balance to the ice-sheet model. At the domain boundaries, the climatological near-surface temperature is imposed, along with desired temperature anomalies. REMBO resolves the snowpack and surface energy balance on the ice-sheet-model grid and resolves the atmospheric dynamics at 120-km resolution. To reduce biases in the simulated present-day ice sheet, an extra 4 m year$^{-1}$ of melt is included in the surface mass balance for areas in which there is no ice present in Greenland today. A simple oceanic anomaly method is used to determine the basal mass balance for marine ice at the grounding line: $\dot{b} = \dot{b}_{ref} + \kappa \Delta T_{ocn}$, in which $\kappa = 10$ m year$^{-1}$ K$^{-1}$ and $\dot{b}_{ref} = -1$ m year$^{-1}$ and $\Delta T_{ocn} = 0.25 T_{2m,ann}$.

Yelmo-REMBO is initialized with the present-day topography and ice-sheet thickness and a semi-analytical solution for the ice-temperature profile at each grid point. The model is then run for 25 kyr to equilibrate the ice sheet with the climatic forcing from REMBO.

This is not long enough to reach full thermodynamic equilibrium, but the ice sheet becomes stable by this point with a well-defined thermodynamic distribution. As with PISM-dEBM, we normalize the ice volume such that the initial volume corresponds to the observed ice volume of 7.42 m sea-level equivalent in all plots[55].

## Climate forcing

The Arctic region is experiencing the most rapid regional warming around the globe[65–67]. To translate the increase in GMT to the warming rate of Greenland and vice versa, we fit the historical (1850–2014) and SSP585 (2015–2100) global mean surface temperature to the mean surface temperature anomaly around Greenland for summer (JJA) from the first available run of the 37 different CMIP6 models to get a scaling factor between regional temperature and GMT increase[46] (Extended Data Table 1). We derive the relationship

$$\Delta GMT_{PI} = f \times \Delta T_{JJA} + 0.5 \text{ °C} \tag{1}$$

between GMT above preindustrial $\Delta GMT_{PI}$ and regional summer temperature increase $\Delta T_{JJA}$ above present. The factor 0.5 °C is the increase of GMT in the reference period for our initial ice sheet states (1980–2000) compared with preindustrial levels (1850–1900) and is derived from HadCRUT5 observational data[68]. The factor $f = \frac{1}{1.19}$ °C$^{-1}$ is the best estimate of the scaling factor between regional Greenland summer temperature and GMT derived from the CMIP6 SSP585 scenarios (Extended Data Table 1)

For the future scenarios, we a apply a spatially constant temperature anomaly with a temperature-dependent seasonal amplitude. We use the scaling factor of 1.61 between regional winter and summer temperature (Extended Data Table 1). We model the difference in the scaling factor between the seasons as a cosine function with a period of 1 year. We fit observational surface temperature in southwestern Greenland for winter and summer from 1850 to 2019 against summer and winter GMT and find consistent scaling factors[68,69] (Extended Data Fig. 8).

## Structural and parametric uncertainties

We address both possible structural and parametric uncertainties of our results. Here structural uncertainties are those associated with the model mechanisms and the structure of the model, whereas parametric uncertainties refer to those that are because of incomplete knowledge of the optimal values for the parameters of a given model.

We account for structural uncertainties by carrying out our experiments with two independent ice-sheet models, PISM-dEBM and Yelmo-REMBO. We show all our results obtained with both models side by side in the figures and conclude that our results are remarkably robust for both models; they are thus unlikely to be affected by structural uncertainties in general, although important differences do arise in the details.

Also, we investigate the parametric uncertainties potentially associated with our results by performing further sensitivity analyses with PISM-dEBM, varying critical parameters that influence the ice dynamics, surface mass balance and further climatic factors (Extended Data Fig. 6). Specifically, we vary the pseudo-plastic sliding exponent, the SSA enhancement factor, the parameter for the bed viscosity, the SIA enhancement factor, the grid resolution, the melt equation parameterization and the precipitation–temperature scaling. Furthermore, we show results without the Earth deformation model.

Although the exact ice-volume loss differs slightly for each combination of the parameters, the qualitative behaviour remains the same. Only the simulation without an Earth deformation model shows a qualitatively different behaviour without a recovery of the ice sheet after an initial loss for some temperature anomalies. This is because of the missing glacial isostatic adjustment. The critical threshold of the ice sheet is not greatly influenced by the ice dynamics parameterization. The melt equation parameterization and

precipitation scaling influence the critical temperature threshold to some extent, yet within the range set by the two independent models. However, the qualitative behaviour does not change and a recovery after an initial loss is seen for all combinations for small temperature anomalies.

It should be noted that, in both models, the ice-sheet response is very sensitive when temperatures are close to the critical thresholds. For example, two simulations with PISM-dEBM show an ice-free state at the temperature of $\Delta T_{JJA} = 2.2$ °C, although the other simulations show a recovery to a mostly glaciated Greenland (Fig. 2a). Similar behaviour can be observed for Yelmo-REMBO, for which one of the simulations shows delayed ice loss when forced with the threshold temperature $\Delta T_{JJA} = 1.5$ °C, but it eventually transitions to the ice-free state. We attribute this to computational errors that can influence the simulations for temperatures very close to the threshold temperature.

## Data availability

The CMIP6 data are freely distributed and available at https://esgf-node.llnl.gov/search/cmip6/ (ref. 46). The BedMachine v5 data are available at https://nsidc.org/data/IDBMG4/versions/5 (refs. 56,60). The output of the regional climate model MARv3.12 is available at ftp://ftp.climato.be/fettweis/MARv3.12/Greenland/ (ref. 59). The observational temperature HadCRUT5 is available at https://www.metoffice.gov.uk/hadobs/hadcrut5/ (ref. 68). The observational ice-sheet velocity MEaSUREs is available at https://nsidc.org/data/NSIDC-0670/versions/1 (refs. 70,71). The datasets generated and analysed during the current study are available on Zenodo at https://doi.org/10.5281/zenodo.8155423.

## Code availability

PISM is open source and freely distributed on GitHub https://github.com/pism/pism. The ice-sheet model Yelmo is open source and freely distributed on GitHub https://github.com/palma-ice/yelmo. The code for analysis and plotting of the model output, as well as an example script of how to run PISM-dEBM, is available in the same Zenodo repository as the model output at https://doi.org/10.5281/zenodo.8155423.

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

**Acknowledgements** This is TiPES contribution #209; the TiPES ('Tipping Points in the Earth System') project has received funding from the European Union's Horizon 2020 research and innovation programme under grant agreement no. 820970. This work was supported by the UiT Aurora Centre Program, UiT The Arctic University of Norway (2020) and the Research Council of Norway (project number 314570). The simulations with PISM-dEBM were performed on the Fram supercomputer provided by Sigma2 - the National Infrastructure for High Performance Computing and Data Storage in Norway under the projects NN8008K and NN9348K. N.Boe. acknowledges further funding by the Volkswagen Foundation and the European Union's Horizon 2020 research and innovation programme under the Marie Skłodowska-Curie grant agreement no. 956170, as well as from the German Federal Ministry of Education and Research under grant no. 01LS2001A. A.R. received funding from the European Union (ERC, FORCLIMA, 101044247). Development of PISM is supported by NSF grants PLR-1644277 and PLR-1914668 and NASA grants NNX17AG65G and 20-CRYO2020-0052. We thank M. Zeitz for her comments on PISM-dEBM that resolved computational problems. Some of the plots are made using scientific colour maps by Crameri et al.[72].

**Author contributions** N.Boe. conceived the study. N.Boc. and N.Boe. designed the study. A.R. performed the experiments with Yelmo-REMBO. N.Boc. performed the experiments with PISM-dEBM and analysed the output data. A.P. assembled and analysed CMIP6 data and wrote parts of the extended data. N.Boc., N.Boe., A.R., M.M. and M.R. discussed and interpreted results. N.Boc. wrote the paper, with contributions from N.Boe., A.R., M.M. and M.R.

**Competing interests** The authors declare no competing interests.

**Additional information**
**Correspondence and requests for materials** should be addressed to Nils Bochow.

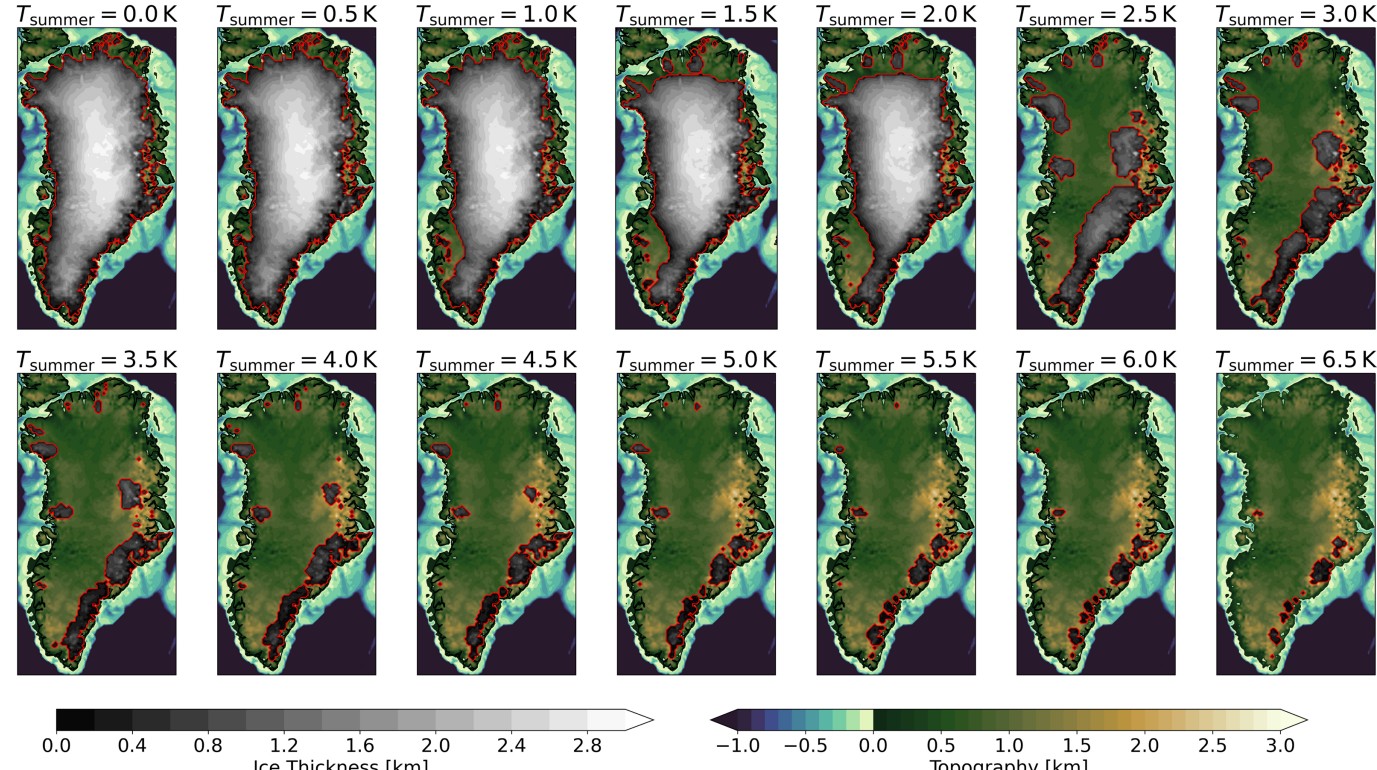

**Extended Data Fig. 1 | Spatial maps of the GrIS after 100 kyr for warming scenarios without mitigation for PISM-dEBM.** Equilibrium states of the GrIS for regional summer warming convergence temperatures between 0 °C and 6.5 °C. The warming period lasts for 100 years and the temperature remains constant afterwards. Several different states can be distinguished: present-day configuration with fully extended ice sheet, several intermediate states with around 50–90% of the present-day ice volume for warming levels between 0 °C and 2.0 °C and an ice-free state. The ice-sheet extent is denoted by a red outline. The spatial configurations correspond to the end states in Fig. 1 and Extended Data Fig. 5. The maps were made with the Python package cartopy[52] and Natural Earth.

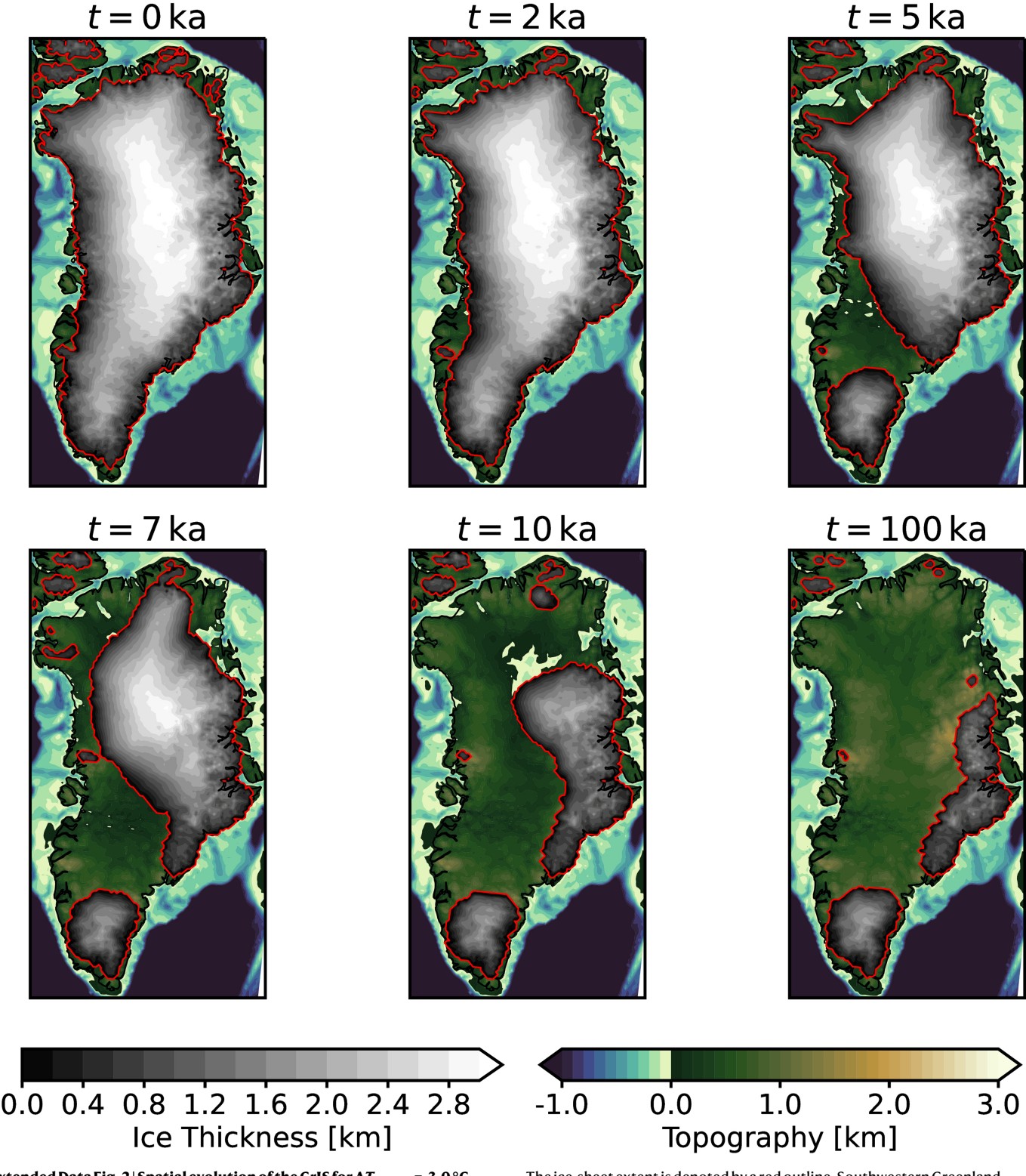

**Extended Data Fig. 2 | Spatial evolution of the GrIS for $\Delta T_{\text{conv,JJA}}$ = 3.0 °C in Yelmo-REMBO.** Exemplary transient snapshots of the GrIS for a regional summer warming convergence temperature of 3.0 °C. The warming period lasts for 100 years and the temperature remains constant afterwards.

The ice-sheet extent is denoted by a red outline. Southwestern Greenland shows the highest sensitivity to warming, followed by the northern part of Greenland. After 10,000 years, most of the ice sheet has melted. The maps were made with the Python package cartopy[53] and Natural Earth.

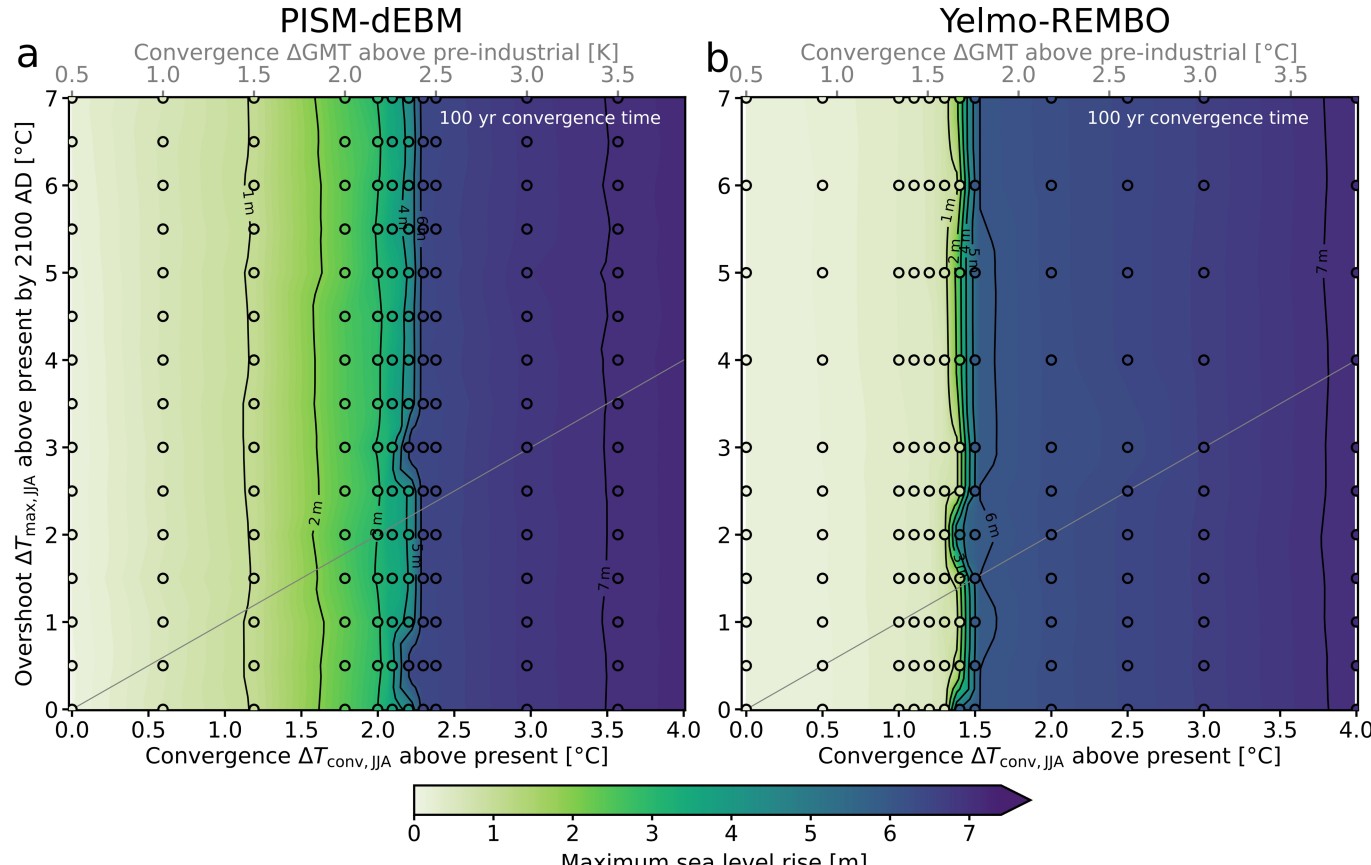

**Extended Data Fig. 3 | Maximum SLR contribution of the GrIS after warming and subsequent cooling for 100 years convergence time. a**, Evolution of ice volume of the whole GrIS for regional summer temperature changes between 0 °C and 7.0 °C above present for PISM-dEBM. The warming period lasts for 100 years, with subsequent cooling for another 100 years to the convergence temperature. Three different states can be distinguished: present-day configuration with fully extended ice sheet, intermediate state with around 60% of the present-day ice volume and an ice-free state. The semi-stable state recovers close to present-day ice-sheet volume after 100 kyr owing to glacial isostatic adjustment. Some runs show oscillatory behaviour on the timescale of several 10 kyr. The corresponding spatial maps are shown in Fig. 1 and Extended Data Fig. 1 and the resulting stability diagram is shown in Fig. 2. **b**, Same as **a** but for Yelmo-REMBO. Only two states are found; present-day and a near-ice-free state.

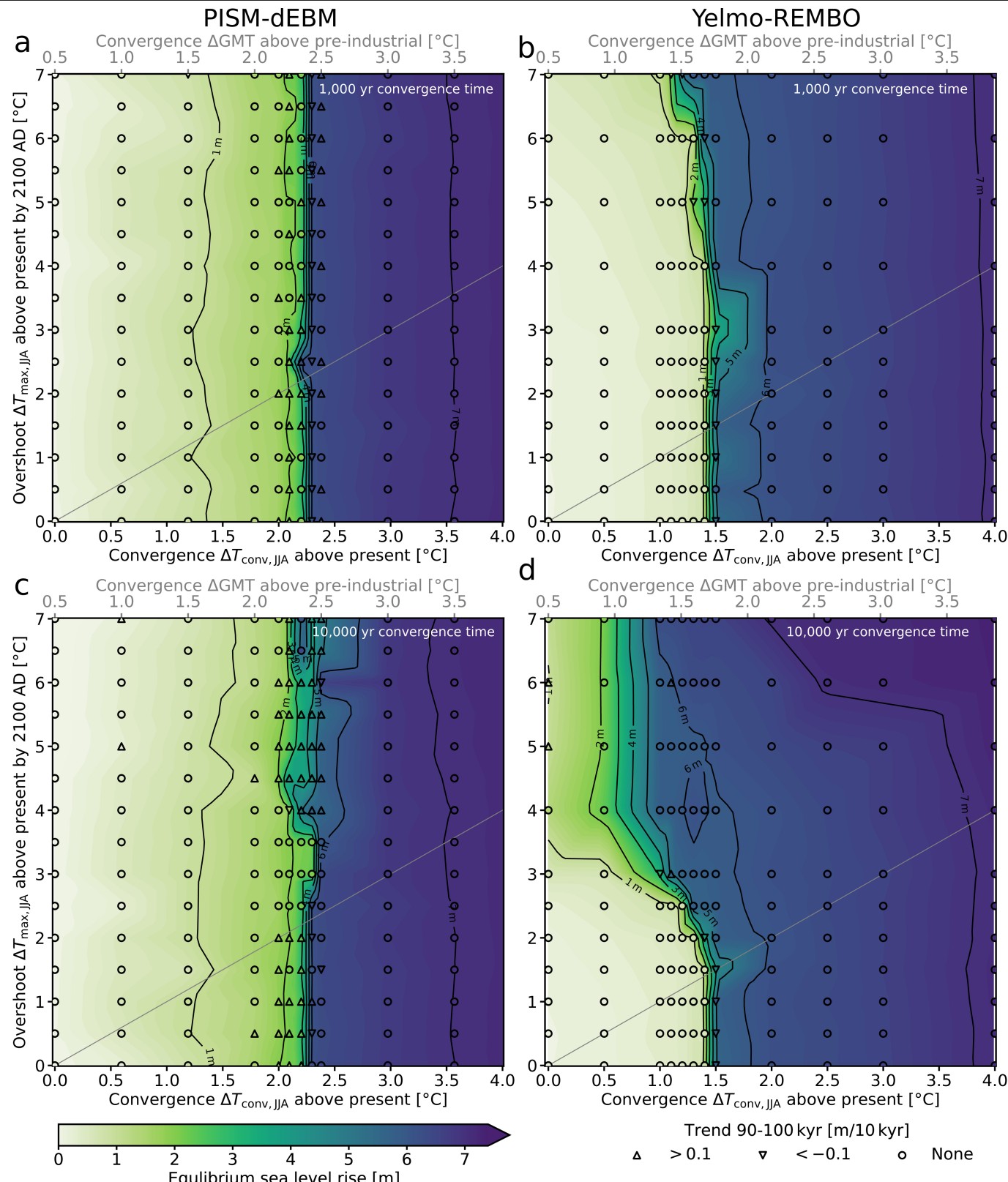

**Extended Data Fig. 4 | Equilibrium SLR contribution of the GrIS after warming and subsequent cooling for convergence times of 1,000 and 10,000 years. a**, Stability diagram of the GrIS with PISM-dEBM. Different warming rates are applied for 100 years, followed by various cooling rates for another 1,000 years. The temperature is kept constant afterwards for another 100 kyr. White regions indicate a present-day-like ice sheet, green–blue regions mark intermediate states and purple corresponds to the ice-free state.

The grey line corresponds to the scenarios for which the overshoot temperature equals the convergence temperature. Below the grey line, the overshoot temperature in the year 2100 AD is smaller than the convergence temperature in 2200 AD. **b**, Same as **a** but for Yelmo-REMBO. **c,d**, Same as **a,b**, respectively, but for a convergence time of 10,000 years. The equilibrium states show a dependence on the overshoot temperature close to the threshold temperature.

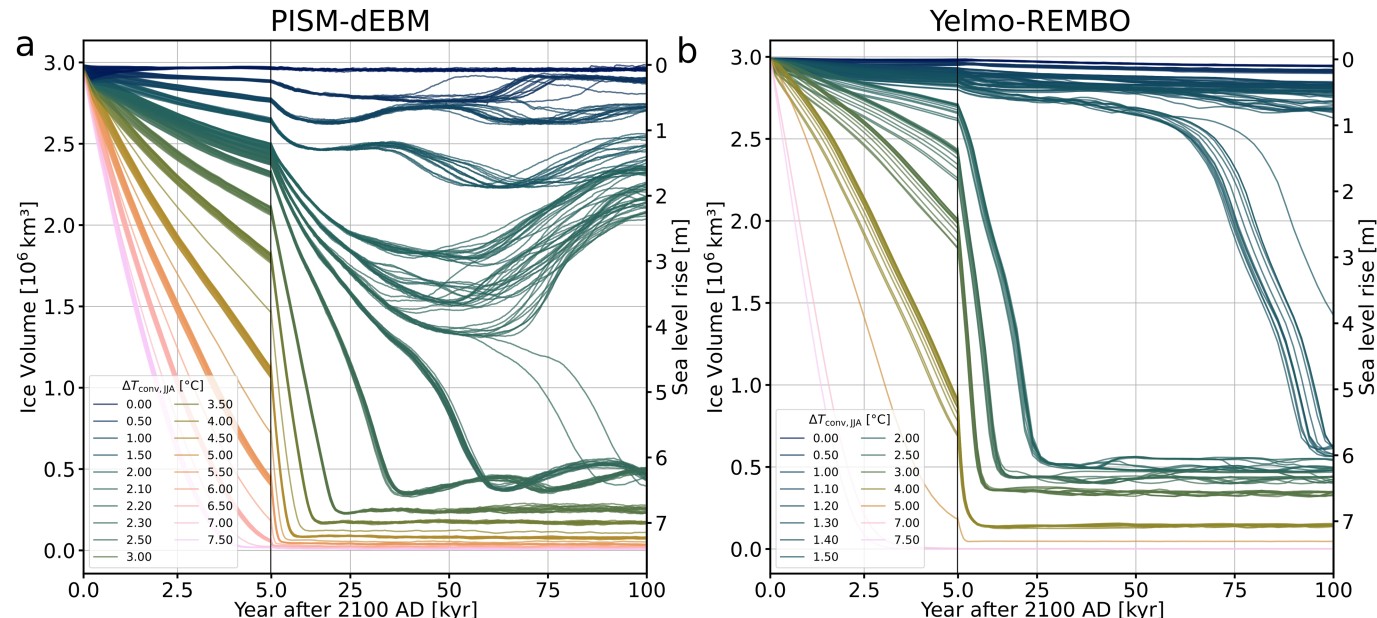

**Extended Data Fig. 5 | Extended time series of ice volume for warming scenarios with mitigation. a**, Evolution of ice volume of the whole GrIS for regional summer temperature changes between 0 °C and 7.5 °C above present for PISM-dEBM. The warming period lasts for 100 years, followed by cooling for another 100 years to the convergence temperature. Three different states are distinguishable: present-day configuration with fully extended ice sheet, intermediate state with around 60% of the present-day ice volume and an ice-free state. The semi-stable state recovers close to the present-day ice-sheet volume after 100 kyr owing to glacial isostatic adjustment. Some runs show oscillatory behaviour on the timescale of several 10 kyr. The corresponding spatial maps are shown in Fig. 1 and Extended Data Fig. 1 and the resulting stability diagram is shown in Fig. 2. **b**, Same as **a** but for Yelmo-REMBO. Only two states are distinguishable; present-day and a near-ice-free state.

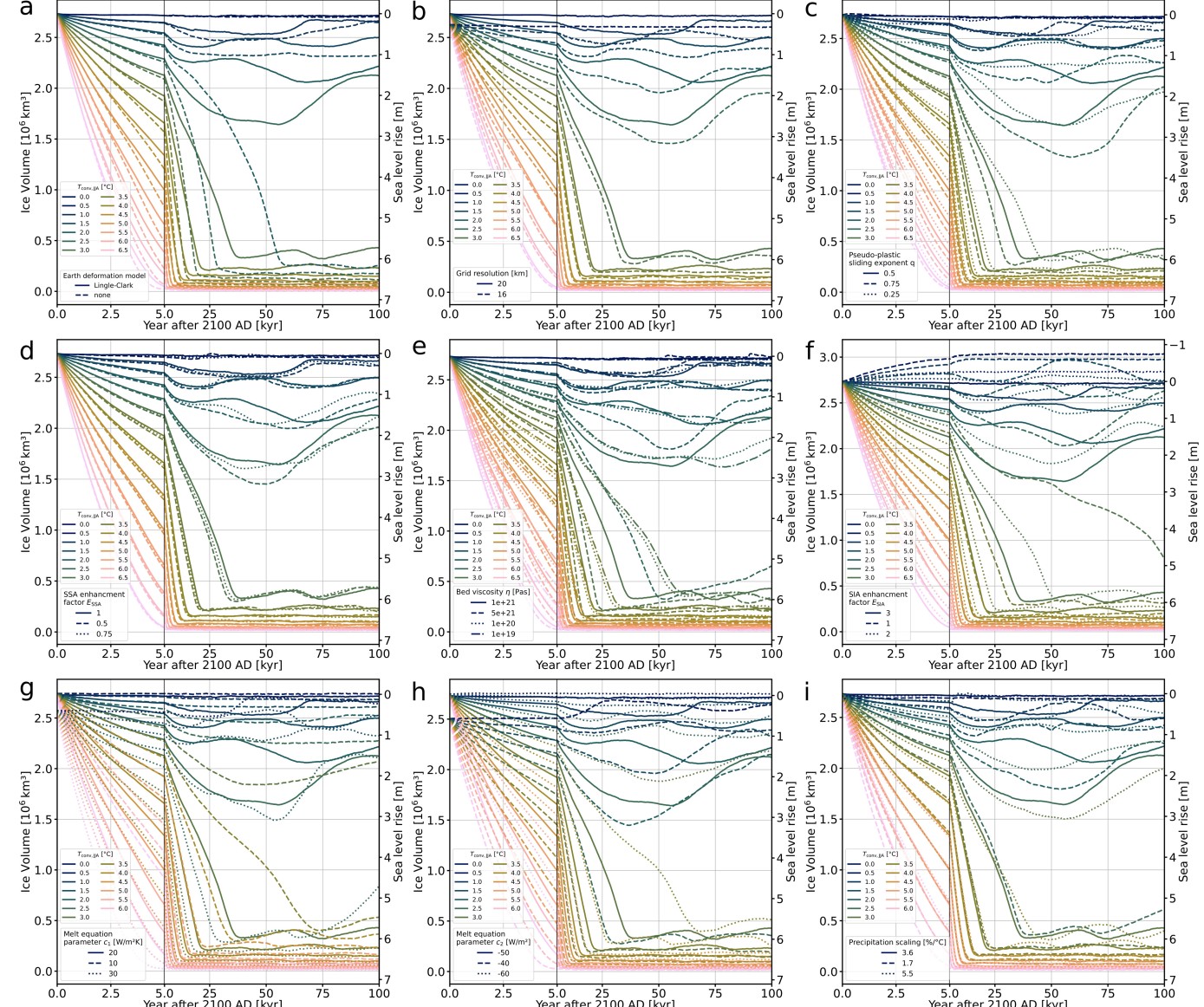

**Extended Data Fig. 6 | Sensitivity of long-term evolution under warming to model parameter variation.** Evolution of total GrIS ice volume simulated by PISM-dEBM, for which the temperature anomalies are not reversed for different temperature anomalies between $\Delta T_{JJA}$ = 0 °C and 6.5 °C above present with variations of important model parameters. The dashed lines correspond to simulations with changes in the parameters compared with the reference parameters. There is no further spin-up of the initial state to account for the parameter changes except for **b**. **a**, Evolution of total GrIS ice volume for regional summer temperature changes between 0 °C and 6.5 °C above present with and without an Earth deformation model. The solid line corresponds to

the reference simulation, as we use it in the main text. **b**–**f**, Same as **a** but for the tested model parameter variation (dashed lines) in comparison with the reference simulation. **b**, Grid resolution. **c**, Pseudo-plastic sliding exponent. **d**, Enhancement factor for the SSA velocities. **e**, Half-space (mantle) viscosity. **f**, Flow enhancement factor for SIA. **g**, Melt equation parameter $c_1$ (see Methods section 'PISM-dEBM'). **h**, Melt equation parameter $c_2$. **i**, Precipitation scaling. Although the exact ice-volume loss differs for the different parameter choices, the qualitative behaviour is the same. Only the simulations without bed deformation model show a qualitatively different behaviour without a recovery after an initial ice loss for temperature anomalies between 1.0 and 2.0 °C.

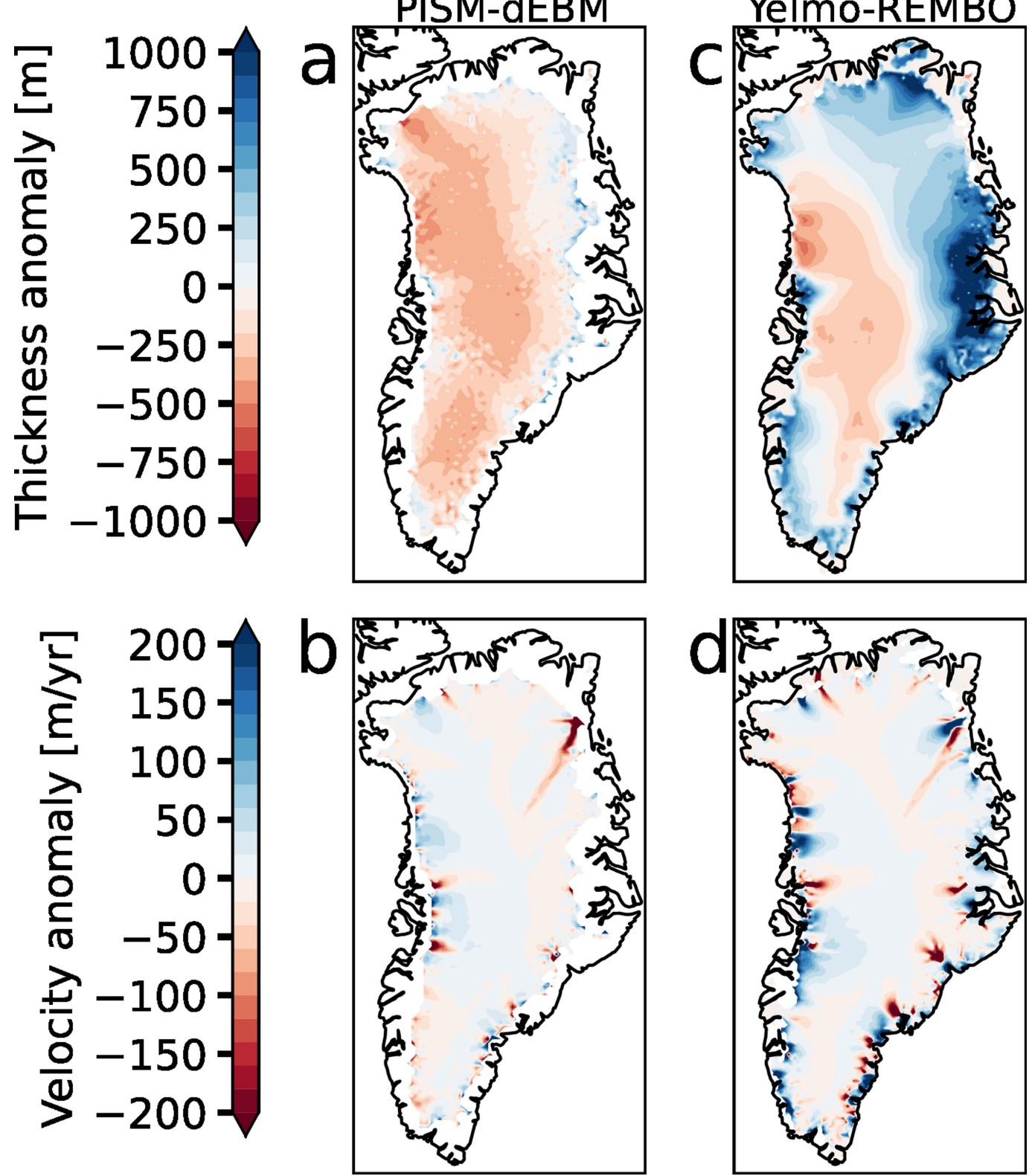

**Extended Data Fig. 7 | Difference between observed and simulated initial-state ice thickness and velocity in PISM-dEBM and Yelmo-REMBO.** **a**, Difference between simulated initial state and observed ice thickness (BedMachine v5 (refs. 56,60)) with PISM-dEBM. Blue and red areas denote regions in which the simulation overestimates and underestimates the thickness, respectively. The root-mean-square error is 260 m. Observational data were regridded to a 20-km grid to ensure comparability. **b**, Same as **a** but for the ice sheet velocity (MEaSUREs v1 (refs. 70,71)). The root-mean-square error is 60 m year⁻¹. **c**,**d**, Same as **a**,**b**, respectively, but for Yelmo-REMBO. The root-mean-square error of the ice thickness is 399 m. The root-mean-square error of the velocity is 83 m year⁻¹. The maps were made with the Python package cartopy[52] and Natural Earth.

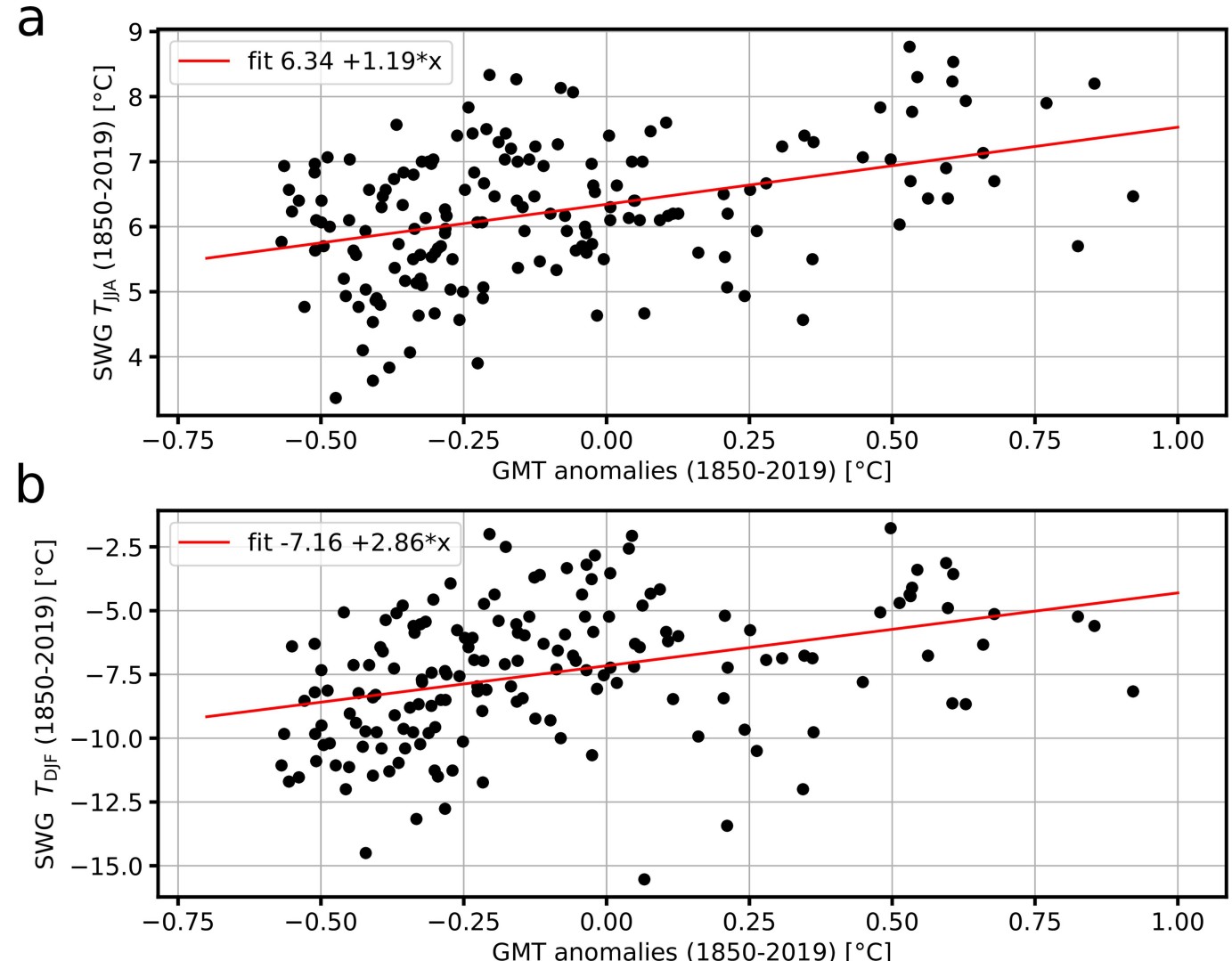

**Extended Data Fig. 8 | Fit of historical air surface temperature in southwestern Greenland against GMT. a**, Linear fit of summer surface air temperature $T_{JJA}$ in southwestern Greenland (SWG)[69] against global mean surface air temperature anomalies for 1850–2019. The GMTs are taken from HadCRUT5 (ref. 68). The scaling factors agree in their uncertainty with the CMIP6-derived scaling factors. **b**, Same as **a** but for winter surface air temperature $T_{DJF}$ in SWG against GMT.

**Extended Data Table 1 | Scaling factors between regional winter surface temperature in Greenland and regional summer temperatures and between regional summer surface temperature and global mean surface temperature**

| Model name | Historical | | SSP585 | |
| --- | --- | --- | --- | --- |
| | DJF/JJA | JJA/GMT | DJF/JJA | JJA/GMT |
| ACCESS-CM2 | 1.24314 | 1.55784 | 1.68832 | 1.10120 |
| ACCESS-ESM1-5 | 1.73007 | 1.76506 | 2.02724 | 1.04376 |
| AWI-CM-1-1-MR | 1.32796 | 1.25945 | 1.82360 | 0.99303 |
| BCC-CSM2-MR | 1.29499 | 0.90418 | 2.26547 | 0.84801 |
| CAMS-CSM1-0 | 2.15478 | 0.63802 | 2.21623 | 0.67288 |
| CanESM5 | 1.76195 | 1.16298 | 2.02841 | 1.08541 |
| CESM2 | 1.21479 | 1.62255 | 1.39828 | 0.91612 |
| CESM2-WACCM | 1.22269 | 1.51080 | 1.59764 | 1.00008 |
| CIESM | 0.59212 | 0.52382 | 1.17684 | 1.85676 |
| CMCC-CM2-SR5 | 1.00370 | 1.37371 | 1.27209 | 1.55352 |
| CMCC-ESM2 | 1.06552 | 1.39911 | 1.45985 | 1.60628 |
| CNRM-CM6-1 | 1.34151 | 2.86455 | 1.40100 | 1.26130 |
| CNRM-CM6-1-HR | 1.26669 | 1.25802 | 1.60006 | 1.17709 |
| CNRM-ESM2-1 | 1.01460 | 2.55458 | 1.36261 | 1.18301 |
| EC-Earth3 | 1.73116 | 2.64813 | 1.04662 | 1.56726 |
| EC-Earth3-CC | 1.96319 | 3.34523 | 1.06750 | 1.89909 |
| EC-Earth3-Veg | 1.73052 | 2.97792 | 1.09530 | 1.70582 |
| EC-Earth3-Veg-LR | 1.53602 | 3.20890 | 1.08838 | 1.83091 |
| FGOALS-f3-L | 1.53602 | 0.88812 | 2.55654 | 0.99939 |
| FGOALS-g3 | 0.67795 | 1.14430 | 1.70377 | 0.79942 |
| FIO-ESM-2-0 | 1.12462 | 1.34199 | 1.51423 | 1.37146 |
| GFDL-ESM4 | 0.59354 | 0.83519 | 1.16744 | 0.97431 |
| HadGEM3-GC31-LL | 0.88461 | 1.22988 | 1.39971 | 1.30237 |
| HadGEM3-GC31-MM | 0.83721 | 1.44161 | 1.28213 | 1.24049 |
| INM-CM4-8 | 1.20071 | 0.86486 | 2.30557 | 1.12200 |
| INM-CM5-0 | 0.76044 | 1.27254 | 1.84016 | 1.06189 |
| IPSL-CM6A-LR | 1.65678 | 1.46502 | 1.44866 | 1.36732 |
| KIOST-ESM | 0.97104 | 0.91166 | 1.53603 | 0.72659 |
| MIROC6 | 0.85111 | 0.72640 | 1.97290 | 1.20458 |
| MIROC-ES2L | 1.15751 | 0.72640 | 1.94393 | 1.04821 |
| MPI-ESM1-2-HR | 1.43216 | 1.32521 | 1.80874 | 1.03589 |
| MPI-ESM1-2-LR | 1.33128 | 1.30071 | 1.51064 | 1.02537 |
| MRI-ESM2-0 | 0.65250 | 1.21552 | 1.65335 | 0.87347 |
| NESM3 | 1.43084 | 1.56126 | 1.78717 | 1.17106 |
| NorESM2-LM | 0.76920 | 0.63713 | 1.61181 | 1.16325 |
| NorESM2-MM | 0.92682 | 1.11325 | 1.48325 | 0.95163 |
| UKESM1-0-LL | 1.43775 | 1.93959 | 1.37611 | 1.37244 |
| Mean | 1.23408 | 1.48748 | 1.60858 | 1.19223 |
| SD | 0.39992 | 0.72539 | 0.37444 | 0.31023 |

List of the 37 CMIP6 models[46] used for the scaling-factor comparison. The second and fourth columns show the scaling factor between the mean winter surface temperature in Greenland and the mean summer surface temperature in Greenland, respectively, for historical and SSP585 runs. The third and fifth columns show the scaling factor between regional summer surface temperature in Greenland and global mean surface temperature.

**Extended Data Table 2 | Scaling factor of annual mean precipitation in Greenland against mean summer surface temperature in Greenland for SSP585 runs**

| Model name | SSP585 scaling factor [%] |
|---|---|
| ACCESS-CM2 | 4.26337 |
| ACCESS-ESM1-5 | 4.30279 |
| AWI-CM-1-1-MR | 4.43382 |
| BCC-CSM2-MR | 3.35385 |
| CAMS-CSM1-0 | 6.30719 |
| CanESM5 | 7.62772 |
| CESM2 | 0.41463 |
| CESM2-WACCM | 1.16871 |
| CIESM | 2.77636 |
| CMCC-CM2-SR5 | 3.52098 |
| CMCC-ESM2 | 4.40673 |
| CNRM-CM6-1 | 3.03457 |
| CNRM-CM6-1-HR | 4.58830 |
| CNRM-ESM2-1 | 2.12144 |
| EC-Earth3 | 2.77394 |
| EC-Earth3-CC | 3.41980 |
| EC-Earth3-Veg | 3.23276 |
| EC-Earth3-Veg-LR | 3.21419 |
| FGOALS-f3-L | 7.25859 |
| FGOALS-g3 | 0.69556 |
| FIO-ESM-2-0 | 2.53140 |
| GFDL-ESM4 | 1.80231 |
| HadGEM3-GC31-LL | 3.08538 |
| HadGEM3-GC31-MM | 3.12996 |
| INM-CM4-8 | 6.86774 |
| INM-CM5-0 | 5.58120 |
| IPSL-CM6A-LR | 4.62000 |
| KIOST-ESM | 2.61953 |
| MIROC6 | 3.34743 |
| MIROC-ES2L | 4.89060 |
| MPI-ESM1-2-HR | 3.06237 |
| MPI-ESM1-2-LR | 6.28110 |
| MRI-ESM2-0 | 0.62048 |
| NESM3 | 6.08938 |
| NorESM2-LM | 0.87454 |
| NorESM2-MM | 0.55605 |
| UKESM1-0-LL | 3.69790 |
| Mean | 3.58305 |
| SD | 1.91032 |

List of the 37 CMIP6 models[46] used for the comparison. The second column shows the percentage change of the mean annual precipitation in Greenland for changing mean summer surface temperature in Greenland for SSP585.