## [Peer Review File · Nature]

Manuscript Title: Overshooting the critical threshold for the Greenland ice sheet

Reviewer Comments & Author Rebuttals

Reviewer Reports on the Initial Version:

Referees' comments:

Referee #1 (Remarks to the Author):

Bochow et al. present numerical experiments of the Greenland ice sheet over various temperature change scenarios in the future. Their scenarios are constructed as an homogeneous temperature increase over Greenland until 2100 followed by a cooling to a target temperature. They explore both the maximal temperature increase and the speed and amplitude of the cooling after 2100. The methodology is sound, although being simple, and the experiments robust. The paper is well written with good looking figures. I do only have minor comments listed below.

Introduction.

I do not understand why there is so many text on the Greenland ice sheet melt influx to the ocean and its impact on the Atlantic meridional overturning circulation. The authors do not tackle this question in the rest of the manuscript so I found it weird to emphasize this point in the introduction.

Methodology.

The authors use a regional temperature scaling from a global mean temperature change. This sounds reasonable but it is also a very simplified view of Greenland climate change. Apart from polar amplification there might be local temperature change within the Greenland region that an homogeneous scaling will not capture. Perhaps linked for example to elevation. Also, there might be some atmospheric circulation driven regional change, e.g. for precipitation, not captured by the authors' scaling. These things are not easily quantified but they certainly deserve more discussion in the manuscript.

Somehow related to this: for large ice sheet retreat we expect some Atlantic circulation changes, mentioned by the authors in the introduction. For a given global mean temperature, the Greenland climate (temperature and precipitation) will be certainly different with and without an active Atlantic meridional circulation. There might be some EMIC or GCM model outputs available in the literature to quantify this.

I do not fully agree to say that the SMB used here includes the albedo feedback. The change in albedo is indeed parametrised and taken into account for the ice melt. However the albedo has also an importance for the local atmospheric temperature change, not only for the melt. And this is not taken into account here. It again somehow relates to the scaling methodology here: it is possible that low ice elevation will be subjected to larger temperature change due to this albedo feedback. The authors use a spatial resolution of 20 km. This is too coarse to capture any ice sheet ocean interactions. Since ocean driven retreat could be at play in Greenland for some major outlet glaciers, this point may deserve a discussion as it is a limitation of the study.

Discussion.

A better literature review here is needed in terms of threshold temperature for the Greenland ice sheet. At least Pattyn et al. (2018) should be included but there are other papers relevant to put the results in a broader context. Some papers are already cited in the introduction but it would be nice to come back to it in the discussion.

Pattyn, F., Ritz, C., Hanna, E. et al. The Greenland and Antarctic ice sheets under 1.5 °C global warming. *Nature Clim Change* 8, 1053–1061 (2018). <https://doi.org/10.1038/s41558-018-0305-8>

Technical.

P1 L23-24 GMT is spelled out in the second occurrence.

P2 L45,L46,L48 "urgent need", "essential", "paramount". Not sure that this choice of words (subjective) is adequate for a scientific paper.

P3 L75 remove "times"

P6 L149 "not very strongly" -> weakly?

P6 L158 which positive feedbacks? The altitude-melt feedback? Others? Be more precise.

P8 L213 reference?

P9 (b) is showing ice thickness, not elevation as mentioned in the caption.

P11 "State (30-40 ka)" is undefined. Is this a mean over 10 kyr?

P18 phi is used both for the till friction angle for the basal friction and for the elevation angle for the surface mass balance.

P19 L353 what is the value of the time step mentioned here?

P21 L406 where does this number come from? It seems rather low. Does this value have an impact on the simulated ice sheet?

Referee #2 (Remarks to the Author):

This study investigates the potential melting of the Greenland ice sheet under a large range of warming (up till 2100) and cooling thereafter scenarios using a dynamic ice sheet model. This is a very relevant and well-performed study, which can serve as basis for follow-up studies with different ice sheet models and under more settings to eventually come closer to estimate strong thresholds and/or overshoot scenarios.

The authors find an intermediate-ice-cover state for Greenland, which is unclear whether really stable or not. It would be interesting to consider the results of the Pliocene ice sheet model intercomparison project (PLISMIP) on these intermediate ice sheet states.

The manuscript is well written in general. Below I summarize a few comments mostly on clarity and understanding:

Line 75: 'times' is too much.

Line 79/80/81: Consider rephrasing the description of warming/cooling strategy. I had a hard time to understand what has been done. And later, you also use longer cooling periods.

Line 94: what do you mean with states? Real equilibria? Not all of the intermediate states are probably equilibria, but some may be....

Line 147: '... convergence time is not linear' what does this mean here?

Figure 3a: on the y-axis it says min-warming and max-warming. Would be good to include numbers here.

Figure 3b/c: I suppose the dots represent the end state (after 30-40kyr). Why is there always a jump towards this end state at the end of each 'cycle' in the graph?

Figure 4a: The black boundary curves (safe) are very difficult to see.

Referee #3 (Remarks to the Author):

A. Summary of the key results

The paper examines the stability of the Greenland ice sheet in the context of global temperature changes using the PISM ice sheet model. The study is unique in examining the response of the ice sheet to a peak in global temperature, followed by a temperature decrease to a lower value that could result from atmospheric removal of CO₂ for example. The study finds that in general the ice sheet response is sensitive to the equilibrium temperature following the perturbation (convergence temperature), rather than the magnitude of the initial temperature increase. Above a final global temperature value of $\sim 4.5^{\circ}\text{C}$ however, the ice sheet is ultimately lost. The response is also sensitive to the timing of the subsequent temperature reduction, with a longer time to decrease temperature resulting in additional sea level rise, before a subsequent stabilization. Despite the ability of a return to a lower temperature anomaly serving to stabilize the ice sheet, in the interim between the temperature maximum and the stabilization temperature, there can be considerable sea level rise, with the amount depending on the temperature peak, the convergence temperature, and the time it takes to reach the convergence temperature.

B. Originality and significance: if not novel, please include reference

The study is novel in that it examines the overshooting and subsequent temperature reduction scenarios. To my knowledge this has not been investigated before. The study is significant because it implies that even if an apparently irreversible temperature threshold for ice sheet loss is reached, it is still possible to prevent the loss of an ice sheet by subsequently reducing global temperatures to a more manageable convergence value. It also indicates that the sea level rise risk can be mitigated by reaching the convergence value more rapidly.

C. Data & methodology: validity of approach, quality of data, quality of presentation

I feel that the approach is quite interesting and provides important suggestions about how the Greenland ice sheet may behave in response to future climate scenarios. The results are overall clear and the presentation is effective. However, I am mainly concerned with the use of a single ice sheet

model uncoupled from simulation of the rest of the earth system to draw conclusions about future ice sheet evolution. There is little discussion either in the main text or the methods section regarding the ability of PISM to simulate the present-day and paleo evolution of the GrIS. The authors also do not examine the uncertainty that may result from the various assumptions and simplifications that are made, for example regarding surface mass balance processes (e.g. rain vs. snow precipitation, meltwater storage and refreezing in firn, albedo feedbacks associated with uncertainties), and any biases present in dynamical simulation with PISM. Recent studies have utilized ice sheet models to project future changes but reveal considerable differences in ice sheet model simulations. (e.g. Seroussi et al., 2020; Goelzer et al., 2020). It is not clear if similar results would be achieved using one or more models other than PISM. It would also be helpful if the authors state some of the features of the model simulations i.e. which feedbacks are captured, in the main text.

Regarding the presentation, I feel that readers or the public may assume that the simulations are realistic scenarios for the future, but I feel they are more idealistic in mainly examining the ice sheet response to a fixed global mean temperature change, without taking the complexities of the global earth system into account, and some of the feedbacks present on the ice sheet. On 100,000-year timescales there may be considerable changes in global climate, atmospheric and ocean circulation, which could affect the evolution of the system, imposing influences from natural variability that are not part of the climate change response. The authors mention in the methods that orbital variability is taken into account but it is not clear if this is applied in the simulations performed here, and what effect it has. These potential effects and the meaning of the simulations (i.e. whether they are just examining ice evolution or represent plausible future scenarios) should be clarified.

D. Appropriate use of statistics and treatment of uncertainties

I feel that more could be done to quantify the uncertainty associated with the assumptions that go into the simulations performed here. This includes uncertainty in the ice sheet model simulations, the surface mass balance forcing, the relationship between Greenland and global temperatures, feedbacks involving parameterizations of albedo, meltwater and precipitation, the solid earth rebound, and assumptions about ice properties and bedrock characteristics that may affect the ice evolution.

E. Conclusions: robustness, validity, reliability

Again, I feel that the conclusions may be somewhat dependent on the model used and the assumptions made. The results provide an interesting analysis of how the ice could respond to global temperature anomalies, but do not necessarily provide an indication of how the system might actually evolve in the future. I feel the authors have to do more to qualify the validity of their results in the context of other ice sheet models and the various assumptions that have been made. The ice sheet response also seems to be a fairly simple function of the final convergence temperature, and I am wondering if these results are realistic.

F. Suggested improvements: experiments, data for possible revision

I suggest including one or more additional ice sheet models if possible (not necessarily conducting

the full suite of simulations) to examine the robustness of the results. Additionally I feel the authors should attempt to quantify the uncertainty associated with the various assumptions made and assess the impact on the results.

G. References: appropriate credit to previous work?

Some recent studies that discuss modeling of future ice sheet evolution are not mentioned in the study, for example Goelzer et al. (2020), Edwards et al. (2021), Serroussi et al., (2020):

Goelzer, H. et al. (2020) The future sea-level contribution of the Greenland ice sheet: a multi-model ensemble study of ISMIP6, *The Cryosphere*, 14, 3071-3096.

Serroussi, H., et al. (2020) ISMIP6 Antarctica: a multi-model ensemble of the Antarctic ice sheet evolution over the 21st century. *The Cryosphere*, 14, 3033-3070.

Edwards, T., et al. (2021) Projected land ice contributions to twenty-first-century sea level rise. *Nature*, 593, 74-82.

H. Clarity and context: lucidity of abstract/summary, appropriateness of abstract, introduction and conclusions

Overall the writing, presentation and conclusions are clear. The transition between the results and discussion section was a bit abrupt and the topic of isostatic adjustment seemed out of place with regard to the rest of the manuscript.

I feel that care should also be taken to ensure that the results are not taken out of context and to clarify that the actual evolution of the ice sheet could be different from what is simulated here.

Reviewer 1:

Bochow et al. present numerical experiments of the Greenland ice sheet over various temperature change scenarios in the future. Their scenarios are constructed as an homogeneous temperature increase over Greenland until 2100 followed by a cooling to a target temperature. They explore both the maximal temperature increase and the speed and amplitude of the cooling after 2100. The methodology is sound, although being simple, and the experiments robust. The paper is well written with good looking figures. I do only have minor comments listed below.

Thank you for the accurate summary of our study and the overall positive assessment!

Introduction.

I do not understand why there is so many text on the Greenland ice sheet melt influx to the ocean and its impact on the Atlantic meridional overturning circulation. The authors do not tackle this question in the rest of the manuscript so I found it weird to emphasize this point in the introduction.

We think that among the possible impacts of accelerating GrIS melt, North Atlantic freshening is, due to the thereby increasing risk of AMOC slowing, one of the most concerning ones. We agree, however, that we might have put too much emphasis on this and have revised the introduction accordingly.

Methodology.

The authors use a regional temperature scaling from a global mean temperature change. This sounds reasonable but it is also a very simplified view of Greenland climate change. Apart from polar amplification there might be local temperature change within the Greenland region that an homogeneous scaling will not capture. Perhaps linked for example to elevation. Also, there might be some atmospheric circulation driven regional change, e.g. for precipitation, not captured by the authors' scaling. These things are not easily quantified but they certainly deserve more discussion in the manuscript.

Somehow related to this: for large ice sheet retreat we expect some Atlantic circulation changes, mentioned by the authors in the introduction. For a given global mean temperature, the Greenland climate (temperature and precipitation) will be certainly different with and without an active Atlantic meridional circulation. There might be some EMIC or GCM model outputs available in the literature to quantify this.

We agree with the referee regarding the simplifications we made. Concerning the scaling, in the revised manuscript, we have expanded our explanation of our choice of the scaling factor but also highlighted potential caveats of this approach. We also included a short discussion of the effect of an AMOC weakening in the discussion now. In addition, we emphasized again that the climate could significantly change on the time scales we investigate. Please note, however, that the qualitative results of our idealized temperature increase and decrease experiments would not be strongly affected by these additional factors (although they would, of course, be effects quantitatively).

I do not fully agree to say that the SMB used here includes the albedo feedback. The change in albedo is indeed parametrised and taken into account for the ice melt. However the albedo has also an importance for the local atmospheric temperature change, not only for the melt. And this is not taken into account here. It again somehow relates to the scaling methodology here: it is possible that low ice elevation will be subjected to larger temperature change due to this albedo feedback.

The reviewer is correct. In the dEBM snowpack model, only the albedo-melt feedback is explicitly treated, in that a reduced albedo will induce higher melt rates. This is arguably the most important feedback to incorporate. However, the additional atmospheric warming that can result from reducing albedo is missing from this model setup. To some extent this could make these results more conservative. In the revised manuscript, we now present results with a second modeling approach, namely the ice-sheet model Yelmo (similar in physics to PISM) coupled to the simple regional climate model REMBO. In this model framework, both of these feedbacks are accounted for, as the atmosphere is dynamically coupled to the snowpack energy balance. REMBO includes a dynamic atmosphere and, for example, a potential negative feedback as well due to increased precipitation following the retreating ice-sheet margin. Furthermore, the surface energy-balance model used is formulated differently than dEBM, etc. So, it is not possible to easily determine the impact of the different mechanisms in a realistic setting. Nonetheless, we think that by including two different modeling approaches at the state of the art, but with varying degrees of complexity, we can be confident in the results, which are quite consistent between them.

The authors use a spatial resolution of 20 km. This is too coarse to capture any ice sheet ocean interactions. Since ocean driven retreat could be at play in Greenland for some major outlet glaciers, this point may deserve a discussion as it is a limitation of the study.

Thank you for pointing this out. We show some results with higher resolution (16 km) with PISM-dEBM in the SI and show that the higher resolution does not change the qualitative results. Furthermore, we include an additional ice sheet model (Yelmo), which we run at 16 km resolution. Even higher resolution would be needed to capture ice-ocean interactions, but this was not feasible on the timescales of interest. We now include a short discussion about the implications of resolution choice.

Discussion.

A better literature review here is needed in terms of threshold temperature for the Greenland ice sheet. At least Pattyn et al. (2018) should be included but there are other papers relevant to put the results in a broader context. Some papers are already cited in the introduction but it would be nice to come back to it in the discussion.

Pattyn, F., Ritz, C., Hanna, E. et al. The Greenland and Antarctic ice sheets under 1.5 °C global warming. *Nature Clim Change* 8, 1053–1061 (2018). <https://doi.org/10.1038/s41558-018-0305-8>

The referee is right; we have included Pattyn et al. but also include a few more important studies in the revised version, and also put our results into context in the discussion now.

Technical.

P1 L23-24 GMT is spelled out in the second occurrence.

Thank you for this careful observation, which we have now fixed.

P2 L45,L46,L48 "urgent need", "essential", "paramount". Not sure that this choice of words (subjective) is adequate for a scientific paper.

We agree and have revised our manuscript accordingly.

P3 L75 remove "times"

Thanks, this has been fixed now.

P6 L149 "not very strongly" -> weakly?

Corrected, thank you.

P6 L158 which positive feedbacks? The altitude-melt feedback? Others? Be more precise.

Thanks, we have now specified this in the revision.

P8 L213 reference?

This sentence has been removed in the revision.

P9 (b) is showing ice thickness, not elevation as mentioned in the caption.

Thank you, we fixed it.

P11 "State (30-40 ka)" is undefined. Is this a mean over 10 kyr?

Thank you for pointing this out. We have modified the legend and we have clarified this in the revised captions.

P18 phi is used both for the till friction angle for the basal friction and for the elevation angle for the surface mass balance.

Thanks, we have changed it to φ in the section on the dEBM.

P19 L353 what is the value of the time step mentioned here?

We have clarified this in the revised manuscript.

P21 L406 where does this number come from? It seems rather low. Does this value have an impact on the simulated ice sheet?

This is the default PISM settings but to be sure we have checked that the specific choice does not influence our simulations.

Reviewer 2:

This study investigates the potential melting of the Greenland ice sheet under a large range of warming (up till 2100) and cooling thereafter scenarios using a dynamic ice sheet model. This is a very relevant and well-performed study, which can serve as basis for follow-up studies with different ice sheet models and under more settings to eventually come closer to estimate strong thresholds and/or overshoot scenarios.

The authors find an intermediate-ice-cover state for Greenland, which is unclear whether really stable or not. It would be interesting to consider the results of the Pliocene ice sheet model intercomparison project (PLISMIP) on these intermediate ice sheet states.

The manuscript is well written in general. Below I summarize a few comments mostly on clarity and understanding:

Thank you for the thorough and overall positive evaluation of our manuscript! In particular, thank you very much for pointing us to PLISMIP. Their results are indeed very interesting, not only because of the intermediate states, but also since they observe similar oscillations as we see in PISM. We mention this paper now in the discussion.

Line 75: 'times' is too much.

Thanks, we fixed this.

Line 79/80/81: Consider rephrasing the description of warming/cooling strategy. I had a hard time to understand what has been done. And later, you also use longer cooling periods.

Yes we agree this was not very clear and have revised it, hoping that it is clearer now.

Line 94: what do you mean with states? Real equilibria? Not all of the intermediate states are probably equilibria, but some may be...

We clarified that not all "states" reach an equilibrium but rather oscillate.

Line 147: '... convergence time is not linear' what does this mean here?

Thanks for the careful evaluation here, we have clarified this in the revised manuscript.

Figure 3a: on the y-axis it says min-warming and max-warming. Would be good to include numbers here.

This schematic is now part of Figure 1; note that we use different maximum temperatures and convergence times in our experiments and we would hence like to keep the "T_max" and "T_conv" on the y-axis.

Figure 3b/c: I suppose the dots represent the end state (after 30-40kyr). Why is there always a jump towards this end state at the end of each 'cycle' in the graph?

Yes, exactly. The reason is that the ice sheet did not necessarily already reach its equilibrium for the new convergence temperature for short convergence times. We have clarified this in the caption and main text.

Figure 4a: The black boundary curves (safe) are very difficult to see.

We do not include this figure anymore, but there are similar ones. We changed the colormap and the contours, which should be easier to read now. Thanks a lot for pointing this out!

Reviewer 3:

A. Summary of the key results

The paper examines the stability of the Greenland ice sheet in the context of global temperature changes using the PISM ice sheet model. The study is unique in examining the response of the ice sheet to a peak in global temperature, followed by a temperature decrease to a lower value that could result from atmospheric removal of CO₂ for example. The study finds that in general the ice sheet response is sensitive to the equilibrium temperature following the perturbation (convergence temperature), rather than the magnitude of the initial temperature increase. Above a final global temperature value of ~4.5°C however, the ice sheet is ultimately lost. The response is also sensitive to the timing of the subsequent temperature reduction, with a longer time to decrease temperature resulting in additional sea level rise, before a subsequent stabilization. Despite the ability of a return to a lower temperature anomaly serving to stabilize the ice sheet, in the interim between the temperature maximum and the stabilization temperature, there can be considerable sea level rise, with the amount depending on the temperature peak, the convergence temperature, and the time it takes to reach the convergence temperature.

Thank you for this accurate summary of our study.

B. Originality and significance: if not novel, please include reference

The study is novel in that it examines the overshooting and subsequent temperature reduction scenarios. To my knowledge this has not been investigated before. The study is significant because it implies that even if an apparently irreversible temperature threshold for ice sheet loss is reached, it is still possible to prevent the loss of an ice sheet by subsequently reducing global temperatures to a more manageable convergence value. It also indicates that the sea level rise risk can be mitigated by reaching the convergence value more rapidly.

Thank you for this assessment regarding our study's novelty and significance.

C. Data & methodology: validity of approach, quality of data, quality of presentation

I feel that the approach is quite interesting and provides important suggestions about how the Greenland ice sheet may behave in response to future climate scenarios. The results are overall clear and the presentation is effective. However, I am mainly concerned with the use of a single ice sheet model uncoupled from simulation of the rest of the earth system to draw conclusions about future ice sheet evolution. There is little discussion either in the main text or the methods section regarding the ability of PISM to simulate the present-day and paleo evolution of the GrIS. The authors also do not examine the uncertainty that may result from the various assumptions and simplifications that are made, for example regarding surface mass balance processes (e.g. rain vs. snow precipitation, meltwater storage and refreezing in firn, albedo feedbacks associated with uncertainties), and any biases present in dynamical simulation with PISM. Recent studies have utilized ice sheet models to project future changes but reveal considerable differences in ice sheet model simulations. (e.g. Seroussi et

al., 2020; Goelzer et al., 2020). It is not clear if similar results would be achieved using one or more models other than PISM. It would also be helpful if the authors state some of the features of the model simulations i.e. which feedbacks are captured, in the main text.

We agree with the referee that in the original version of our manuscript, we had not paid sufficient attention to structural and parametric uncertainties. To address this, we have repeated our analysis with a second ice sheet model with a different setup, and have also included sensitivity studies regarding variations of key parameters for PISM-dEBM. In our revised manuscript, we now present all results for the two models side by side in each figure, so that readers can obtain a direct feeling of the cross-model uncertainty of our results. Moreover, we have added a methods section on structural and parametric uncertainties. For the latter, we now perform a detailed sensitivity analysis focussing on the effects of varying the pseudo-plastic sliding exponent, the SSA enhancement factor, the parameter for the bed viscosity, the SIA enhancement factor, the grid resolution, the melt equation parameterisation, and the precipitation-temperature scaling. We also investigate the effect of not including the Earth deformation model. We can on this basis show that our results are robust across different models, and that varying the above parameters within reasonable bounds does not qualitatively affect our results; please see Fig. S6

Regarding the presentation, I feel that readers or the public may assume that the simulations are realistic scenarios for the future, but I feel they are more idealistic in mainly examining the ice sheet response to a fixed global mean temperature change, without taking the complexities of the global earth system into account, and some of the feedbacks present on the ice sheet. On 100,000-year timescales there may be considerable changes in global climate, atmospheric and ocean circulation, which could affect the evolution of the system, imposing influences from natural variability that are not part of the climate change response. The authors mention in the methods that orbital variability is taken into account but it is not clear if this is applied in the simulations performed here, and what effect it has. These potential effects and the meaning of the simulations (i.e. whether they are just examining ice evolution or represent plausible future scenarios) should be clarified.

Thank you for this observation! In the revised manuscript we have made it clearer that we consider idealized future scenarios and have also added a short discussion about the uncertainty related to the long-term future of the climate system on the investigated timescales. We also clarified that we keep the orbital parameters fixed in the methods.

D. Appropriate use of statistics and treatment of uncertainties

I feel that more could be done to quantify the uncertainty associated with the assumptions that go into the simulations performed here. This includes uncertainty in the ice sheet model simulations, the surface mass balance forcing, the relationship between Greenland and global temperatures, feedbacks involving parameterizations of albedo, meltwater and precipitation, the solid earth rebound, and assumptions about ice properties and bedrock characteristics that may affect the ice evolution.

We agree and have added extensive uncertainty analyses in the revised manuscript. In particular, we now use two independent ice sheet models and show that our results are consistent. We have also conducted thorough uncertainty analyses concerning the most

relevant parameters in PISM-dEBM, i.e. the pseudo-plastic sliding exponent, the SSA enhancement factor, the parameter for the bed viscosity, the SIA enhancement factor, the grid resolution, the melt equation parameterisation, and the precipitation-temperature scaling (see Fig. S6).

E. Conclusions: robustness, validity, reliability

Again, I feel that the conclusions may be somewhat dependent on the model used and the assumptions made. The results provide an interesting analysis of how the ice could respond to global temperature anomalies, but do not necessarily provide an indication of how the system might actually evolve in the future. I feel the authors have to do more to qualify the validity of their results in the context of other ice sheet models and the various assumptions that have been made. The ice sheet response also seems to be a fairly simple function of the final convergence temperature, and I am wondering if these results are realistic.

We agree that reporting our results from a single ice-sheet model somewhat limited the robustness of the conclusions and have therefore added a second ice-sheet model to our study. We think that with comparing results between two different modeling approaches and the additionally added sensitivity analyses (Fig. S6) in the revised manuscript, our conclusions are much more robust.

F. Suggested improvements: experiments, data for possible revision

I suggest including one or more additional ice sheet models if possible (not necessarily conducting the full suite of simulations) to examine the robustness of the results. Additionally I feel the authors should attempt to quantify the uncertainty associated with the various assumptions made and assess the impact on the results.

As noted above we agree and have indeed performed the entire set of simulations with a second ice sheet model, and have also added extensive uncertainty quantification regarding key parameters of the ice sheet models.

G. References: appropriate credit to previous work?

Some recent studies that discuss modeling of future ice sheet evolution are not mentioned in the study, for example Goelzer et al. (2020), Edwards et al. (2021), Serroussi et al., (2020):
Goelzer, H. et al. (2020) The future sea-level contribution of the Greenland ice sheet: a multi-model ensemble study of ISMIP6, *The Cryosphere*, 14, 3071-3096.
Serroussi, H., et al. (2020) ISMIP6 Antarctica: a multi-model ensemble of the Antarctic ice sheet evolution over the 21st century. *The Cryosphere*, 14, 3033-3070.
Edwards, T., et al. (2021) Projected land ice contributions to twenty-first-century sea level rise. *Nature*, 593, 74-82.

Thank you very much for pointing us to these key references, which we have included in our revised manuscript.

H. Clarity and context: lucidity of abstract/summary, appropriateness of abstract, introduction and conclusions

Overall the writing, presentation and conclusions are clear. The transition between the results and discussion section was a bit abrupt and the topic of isostatic adjustment seemed out of place with regard to the rest of the manuscript.

Thanks, we have revised these parts accordingly.

I feel that care should also be taken to ensure that the results are not taken out of context and to clarify that the actual evolution of the ice sheet could be different from what is simulated here.

We entirely agree and have emphasized even more in the revision that our experiments are idealized, even though we use comprehensive ice sheet models.

Reviewer Reports on the First Revision:

Referees' comments:

Referee #1 (Remarks to the Author):

I appreciate that the authors carefully have taken all my comments into account.

It is also very impressive that they have performed new experiments with an additional ice sheet model (Yelmo). Since PISM and Yelmo includes different sub-model to estimate the surface mass balance, these new experiments considerably strengthen the conclusions.

More specifically, I found particularly interesting the fact that Yelmo does not present intermediate deglaciated states like PISM. I have the feeling that the SMB-ice sheet change feedback is stronger when using REMBO than when using dEBM. It seems to me that it is somehow expected given the fact that REMBO includes more complexity in this feedback. Is this correct? If yes it could be included in the manuscript. Also, would it be possible to broaden a bit this result: would a fully coupled atmospheric model also present only two stable states (glaciated and deglaciated)?

In any case, I would be happy to see the paper published.

Referee #2 (Remarks to the Author):

This is a revised version of a paper I have seen before. All my comments have been adequately addressed in this new version.

I therefore recommend publication.

Referee #3 (Remarks to the Author):

I feel that the manuscript is improved over the previous version and the authors have adequately responded to all reviewer concerns. The addition of the second set of model simulations and sensitivity experiments with one set of model simulations adds confidence in the conclusions and uncertainties associated with the study. I only have some suggested minor corrections listed below:

L. 21-22 It would be helpful if the authors could briefly define overshoot and convergence to give the reader a better understanding of the objectives of the study.

Figure 1: Initially, it was unclear that the top (a) and bottom (b) panels do not correspond to each other. This can be made clearer in the caption. For example, in the text describing (b), the first sentence can mention that (b) shows the model responses to the scenario where the temperature anomaly is not reversed (black line in Figure 1(a)), for different values of ΔT_{\max} . I think it makes more sense to refer to Figure 1a rather than 2a,b in the figure caption, which may be confusing to

the reader.

L. 89-90: Define JJA here; e.g. “linear summer (June, July, August, JJA) temperature increase” and note “maximum summer temperature anomaly” for clarity.

L. 93: Define $\Delta T_{conv, GMT}$.

L. 104: I think this should read “Fig. 1b,f”.

L. 111: I think this should read “Fig. 1c-e”.

L. 120: I think “Fig. 1e” should be changed to “Fig. 1d”.

L. 120: Can the authors briefly explain in the text why glacial isostatic adjustment has this effect? Is it because the surface elevation increases, resulting in a colder climate?

L. 175: Change to “a complete loss of the ice sheet can occur prior to recovery” for clarity.

Fig. 5, caption: Suggest changing to read “Minimum ice volume and maximum sea-level rise contribution” for clarity in the first sentence.

L. 175-179: Again, note that the complete loss is prior to recovery/regrowth.

S6 Caption: Clarify that dashed lines show the results of sensitivity experiments.

L. 245: Clarify the meaning of “reasonable time”.

L. 445: Can the authors note the source of the bed elevation data?

L. 458: MAR hasn’t been defined yet. Revise to “the MAR v3.12 regional climate model surface mass balance from 1980 to 2000 [57],” and revise text on line 463.

L. 478: Change “of or” to “of”.

L. 527: Is the amplitude here the amplitude of interannual variability? Please clarify.

Author Rebuttals to First Revision:

Referee #1:

I appreciate that the authors carefully have taken all my comments into account.

It is also very impressive that they have performed new experiments with an additional ice sheet model (Yelmo). Since PISM and Yelmo includes different sub-model to estimate the surface mass balance, these new experiments considerably strengthen the conclusions.

Thank you for the careful evaluation and the positive assessment of our revised manuscript; your comments on the first version have helped us a lot in further improving the manuscript.

More specifically, I found particularly interesting the fact that Yelmo does not present intermediate deglaciated states like PISM. I have the feeling that the SMB-ice sheet change feedback is stronger when using REMBO than when using dEBM. It seems to me that it is somehow expected given the fact that REMBO includes more complexity in this feedback. Is this correct? If yes it could be included in the manuscript.

This is an interesting point, but one that is difficult to know with confidence. At a minimum, it is clear that the existence or not of intermediate states is model dependent. The additional feedback of atmospheric warming with higher melt also would point to a stronger positive feedback cycle within Yelmo-REMBO. However, the counterbalancing effect of increased slope-driven precipitation in REMBO is also included. Furthermore, the response of ice dynamics with Yelmo compared to PISM can play a large role too on these longer timescales. For this reason, we prefer to avoid putting too much emphasis on REMBO itself as the cause of the lack of intermediate states. Throughout the manuscript, we have tried to be clear about which feedbacks and mechanisms are represented in the different models. And in the Discussion, we simply note that despite their consistency in general concerning overshooting, the existence or not of intermediate states for a simple warming scenario appears to be model dependent (L255-257 in the revised text).

Also, would it be possible to broaden a bit this result: would a fully coupled atmospheric model also present only two stable states (glaciated and deglaciated)?

Thank you for this suggestion. It would certainly be very interesting to investigate this problem with an atmosphere GCM fully coupled to the ice-sheet dynamics, for example following the approach of

Gregory et al. (2020). Yelmo-REMBO does include a dynamic atmosphere that, e.g., produces increased precipitation following the retreating ice-sheet margin. However, this modeling approach cannot capture large-scale changes in atmospheric dynamics that may be relevant to SMB and stability. This would be beyond the scope of this present study, but we note in the revised manuscript that this is an important path for follow-up work.

In any case, I would be happy to see the paper published.

Thank you!

Referee #2:

This is a revised version of a paper I have seen before. All my comments have been adequately addressed in this new version.

I therefore recommend publication.

We are happy that we were able to satisfactorily respond to all concerns raised by Referee #2.

Referee #3:

I feel that the manuscript is improved over the previous version and the authors have adequately responded to all reviewer concerns. The addition of the second set of model simulations and sensitivity experiments with one set of model simulations adds confidence in the conclusions and uncertainties associated with the study.

Thank you, we are glad that we could satisfactorily address your concerns!

I only have some suggested minor corrections listed below:

L. 21-22 It would be helpful if the authors could briefly define overshoot and convergence to give the reader a better understanding of the objectives of the study.

We agree and have added a corresponding sentence in the introduction; thank you for this suggestion!

Figure 1: Initially, it was unclear that the top (a) and bottom (b) panels do not correspond to each other. This can be made clearer in the caption. For example, in the text describing (b), the first sentence can mention that (b) shows the model responses to the scenario where the temperature anomaly is not reversed (black line in Figure 1(a)), for different values of T_{max} . I think it makes more sense to refer to Figure 1a rather than 2a,b in the figure caption, which may be confusing to the reader.

We agree and have modified the caption of Figure 1 accordingly.

L. 89-90: Define JJA here; e.g. “linear summer (June, July, August, JJA) temperature increase” and note “maximum summer temperature anomaly” for clarity.

Thank you! We modified the sentences accordingly.

L. 93: Define $T_{conv,GMT}$.

We now define $T_{conv,GMT}$ and refer to the Climate forcing section in the methods.

L. 104: I think this should read “Fig. 1b,f”.

Indeed! Thank you.

L. 111: I think this should read “Fig. 1c-e”.

Thank you!

L. 120: I think "Fig. 1e" should be changed to "Fig. 1d".

Thank you!

L. 120: Can the authors briefly explain in the text why glacial isostatic adjustment has this effect? Is it because the surface elevation increases, resulting in a colder climate?

We now briefly explain it in these lines: The uplift of the bedrock counteracts the melt-elevation feedback and leads to colder temperatures which allow the ice sheet to partially regrow (Zeitz et al. 2021).

L. 175: Change to "a complete loss of the ice sheet can occur prior to recovery" for clarity.

Thank you for that comment! We now clarify that the loss is prior to a recovery.

Fig. 5, caption: Suggest changing to read "Minimum ice volume and maximum sea-level rise contribution" for clarity in the first sentence.

Thank you! That definitely improves the clarity of that figure.

L. 175-179: Again, note that the complete loss is prior to recovery/regrowth.

Thank you for the comment! We modified the sentence accordingly.

S6 Caption: Clarify that dashed lines show the results of sensitivity experiments.

We agree that it was not clear enough that the dashed lines are sensitivity experiments, we make it clear now in the caption.

L. 245: Clarify the meaning of “reasonable time”.

Indeed, we did not clarify specifically what we mean by reasonable time. We now state it clearly in the text (within centuries).

L. 445: Can the authors note the source of the bed elevation data?

We include the source for the bed elevation data now (BedMachine v5).

L. 458: MAR hasn't been defined yet. Revise to “the MAR v3.12 regional climate model surface mass balance from 1980 to 2000 [57],” and revise text on line 463.

Thank you for that suggestion! We revised the sentence accordingly.

L. 478: Change “of or” to “of”.

Thank you, we corrected it.

L. 527: Is the amplitude here the amplitude of interannual variability? Please clarify.

Here we mean the seasonal amplitude. We now state it clearly.

References:

Gregory, J. M., George, S. E., and Smith, R. S.: Large and irreversible future decline of the Greenland ice sheet, *The Cryosphere*, 14, 4299–4322, <https://doi.org/10.5194/tc-14-4299-2020>, 2020.